# Injury-induced perivascular niche supports alternative differentiation of adult rodent CNS progenitor cells

Justyna Ulanska-Poutanen[1], Jakub Mieczkowski[1], Chao Zhao[2,3], Katarzyna Konarzewska[1], Beata Kaza[1], Hartmut BF Pohl[4], Lukasz Bugajski[5], Bozena Kaminska[1], Robin JM Franklin[2,3]*, Malgorzata Zawadzka[1]*

[1]Laboratory of Molecular Neurobiology, Neurobiology Center, Nencki Institute of Experimental Biology, Polish Academy of Sciences, Warsaw, Poland; [2]Wellcome Trust - Medical Research Council Cambridge Stem Cell Institute, University of Cambridge, Cambridge, United Kingdom; [3]Department of Clinical Neurosciences, University of Cambridge, Cambridge, United Kingdom; [4]Department of Biology, Institute of Molecular Health Sciences, Zurich, Switzerland; [5]Laboratory of Cytometry, Nencki Institute of Experimental Biology, Polish Academy of Sciences, Warsaw, Poland

**Abstract** Following CNS demyelination, oligodendrocyte progenitor cells (OPCs) are able to differentiate into either remyelinating oligodendrocytes (OLs) or remyelinating Schwann cells (SCs). However, the signals that determine which type of remyelinating cell is generated and the underlying mechanisms involved have not been identified. Here, we show that distinctive microenvironments created in discrete niches within demyelinated white matter determine fate decisions of adult OPCs. By comparative transcriptome profiling we demonstrate that an ectopic, injury-induced perivascular niche is enriched with secreted ligands of the BMP and Wnt signalling pathways, produced by activated OPCs and endothelium, whereas reactive astrocyte within non-vascular area express the dual BMP/Wnt antagonist Sostdc1. The balance of BMP/Wnt signalling network is instructive for OPCs to undertake fate decision shortly after their activation: disruption of the OPCs homeostasis during demyelination results in BMP4 upregulation, which, in the absence of Socstdc1, favours SCs differentiation.
DOI: https://doi.org/10.7554/eLife.30325.001

*For correspondence:
rjf1000@cam.ac.uk (RJMF);
m.zawadzka@nencki.gov.pl (MZ)

**Competing interests:** The authors declare that no competing interests exist.

## Introduction

CNS remyelination is mediated by a specialized type of adult multipotent progenitor cells, commonly referred to oligodendrocyte progenitor cells or OPCs, that are able to differentiate into new myelin-forming oligodendrocytes (OLs) following primary demyelination. This regenerative process can also involve the generation of remyelinating Schwann cells (SCs), the myelin-forming cells of the peripheral nervous system (PNS). The contribution of SCs to CNS remyelination have been reported in several clinical conditions including traumatic spinal cord injury, leukodystrophies (*Bunge et al., 1993*; *Guest et al., 2005*), and most frequently in severe forms of multiple sclerosis (MS) (*Itoyama et al., 1983*). Experimentally, this phenomenon has been observed following virtually all forms of CNS injury with a demyelinating component, including ischemic, excitotoxic, and traumatic injury as well as in toxic and immune-mediated models of demyelination (*Snyder et al., 1975*; *Dusart et al., 1992*; *Felts et al., 2005*). It was originally assumed that SCs invaded demyelinated regions of the CNS from the periphery following a breach in the *glia limitans* (for review see *Franklin and Blakemore, 1993*). However, we have previously shown using a genetic fate mapping

strategy that following CNS demyelination, adult OPCs are presented with a fate choice, having the option to become OLs or SCs as they contribute to remyelination (*Zawadzka et al., 2010*). Recently, *Assinck et al., 2017* demonstrated extensive Schwann cell-mediated remyelination following clinically relevant traumatic spinal contusion injury and using genetic reporters provided confirmatory evidence for their central origin.

The underlying mechanism controlling this unusual CNS-to-PNS fate-switching of adult OPCs is unclear. SC-mediated remyelination of central axons is closely associated with localization of cells in the lesion and cellular composition of the surrounding tissue. We and others have reported that SCs can be predominantly found in the CNS areas from which astrocytes are absent (*Woodruff and Franklin, 1999*; *Blakemore, 1975*; *Blakemore, 2005*; *Talbott et al., 2006*). The central role of astrocytes in determining which type of remyelination occurs has recently been demonstrated by increased SCs remyelination when the astrocyte response to demyelination has been reduced either transgenically (*Monteiro de Castro et al., 2015*) or by reducing testosterone signalling (*Bielecki et al., 2016*). Transplantation studies suggest that the molecular composition of an astrocyte-free CNS environment promotes SC differentiation of adult OPCs, possibly via a mechanism that involves bone morphogenetic proteins (BMPs) (*Talbott et al., 2006*). However, BMPs alone are unlikely to induce SC differentiation since they primarily promote OPC differentiation into astrocytes in vivo (*Mabie et al., 1997*; *Grinspan et al., 2000*; *Gomes et al., 2003*; *Cheng et al., 2007*; *Sabo et al., 2011*) or astrocyte and neuronal fate in vitro (*Kondo and Raff, 2004*).

Fate decisions by adult multipotential cells are often regulated by a specialized microenvironment, termed the 'niche', associated with the vasculature (*Goldman and Chen, 2011*). Injury-induced loss of the local vasculature and disruption of blood brain barrier (BBB) integrity is a common pathological feature of demyelinating disease, while tissue reconstruction is associated with enhanced angiogenesis and the reestablishment of a functional vasculature (*Miyamoto et al., 2014*; *Egawa et al., 2016*). We hypothesized that unique properties of the perivascular niche within remyelinating white matter would create microenvironment that favour the alternative differentiation of OPCs. Although the transcriptomic changes associated with OL differentiation have been described (e.g. *Dugas et al., 2006*; *Cahoy et al., 2008*; *Huang et al., 2011*; *Moyon et al., 2015*), the instructive clues and the molecular mechanisms of alternative OPCs differentiation remain unresolved. It is also unclear whether injury-activated endothelium and OPCs may interact with one another during white matter regeneration. We therefore characterised the transcriptomic profile of discrete microenvironmental niches during the early stages of remyelination and identified several factors that significantly discriminate between vascular and non-vascular areas. Our results demonstrate a role of the context-dependent BMP/WNT signalling network in regulation of the alternative, SC differentiation of OPCs.

## Results

### Different cellular compositions define discrete niches within areas of CNS demyelination

To get insight into molecular composition of post-injury niches we used a well-established model of demyelination induced by a stereotactic injection of ethidium bromide (EB) into the caudal cerebellar peduncle (CCP) of adult rats (*Woodruff and Franklin, 1999*). Toxin injection results in focal primary demyelination lesion (*Figure 1A*) with large astrocyte-deficient areas and little axonal loss. The CCP is anatomically remote from spinal root transition zones, which are potential sources of cells of neural crest origin such as SCs or boundary cap cells (*Fraher, 1992*; *Zujovic et al., 2011*).

In response to demyelination, OPCs become activated, proliferate, migrate and ultimately undergo terminal differentiation during several well-defined stages (*Figure 1A*). First, we characterized the spatial distribution of oligodendrocyte lineage cells in the lesion using antibodies to the transcription factor Olig2 and found them preferentially distributed around blood vessels, visualized with agglutinin RCA120, which specifically binds to endothelial cells and their basement membrane (*Figure 1B*). We defined the vascular niche (VN) as an area around blood vessel with a radial length two times greater than a diameter of the particular blood vessel; the remaining lesion tissue was defined as the non-vascular area (non-VN). Numerous Olig2+ cells were preferentially associated with blood vessels and reached their highest density in the VN at 10 days post-lesion induction (dpl)

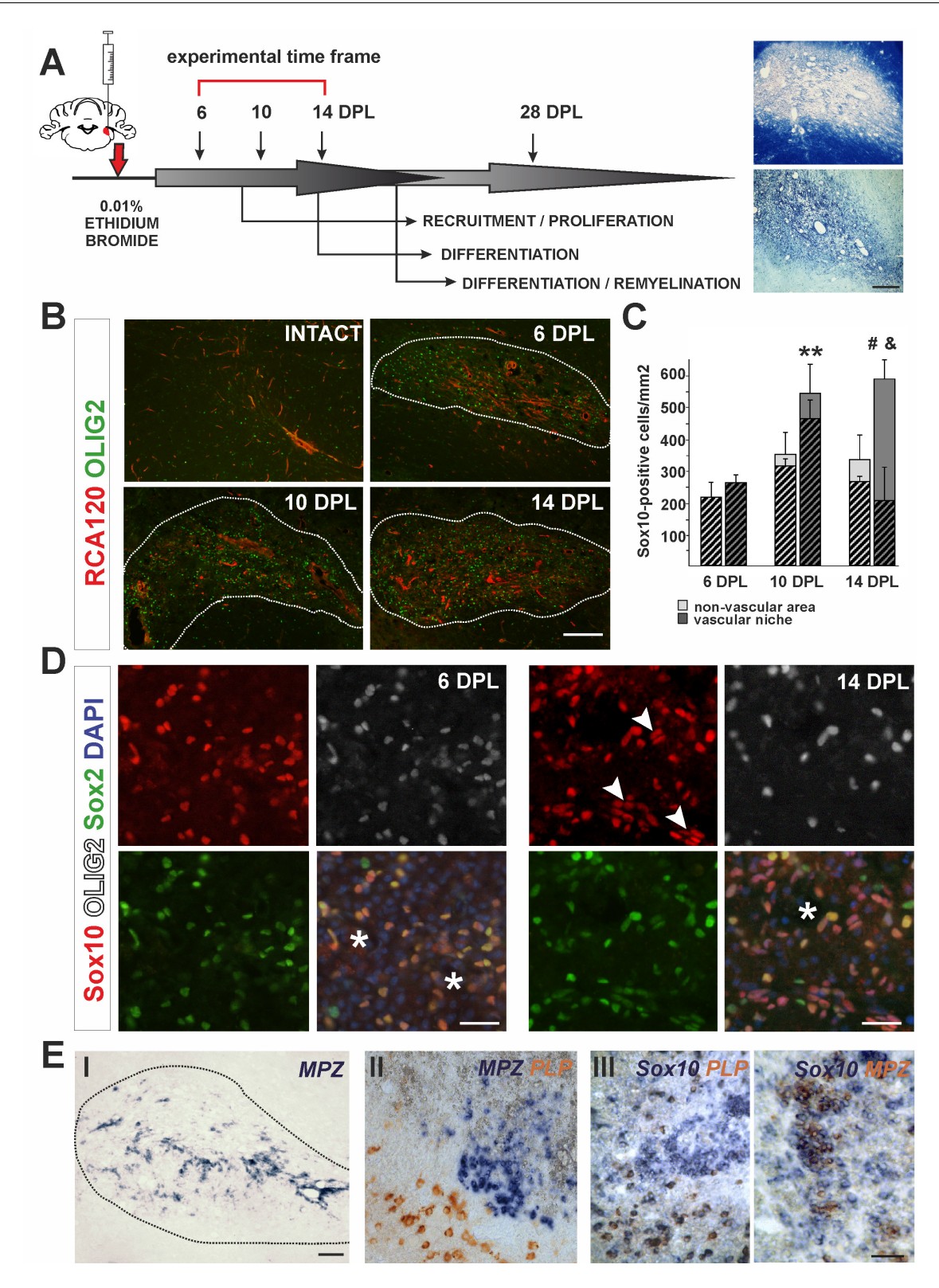

**Figure 1.** The cellular composition of VN differs from non-VN during remyelination. (A) Schematic description of OPCs response to focal demyelination in rat CCP. Note the well-defined phases: recruitment of OPCs, their proliferation and differentiation (6, 10 and 14 dpl). Images show lesion area identified by detection of myelin (upper image) and cellularity (lower image, scale bar 200 µm). (B) Representative images of blood vessels (RCA120, red) and Olig2+ cells (green) distribution after CCP demyelination, scale bar 200 µm, lesion depicted by dotted line. (C) Sox10+ cells quantification,

*Figure 1 continued on next page*

*Figure 1 continued*

Student's t-test; **p<0.01 significant difference between VN at 6 dpl and 10 dpl as well as between VN and non-VN at 10 dpl; #p<0.05, significant difference between VN at 10 dpl and VN at 14 dpl; n = 3; shaded bars show the number of Olig2+ among Sox10+ cells. The proportion of Olig2- cells among Sox10+ cells within VN increases sharply between 10 dpl and 14 dpl (Student's t-test; and p<0.001; n = 3). (D) Sox10+ cells remain Sox2+ regardless the Olig2 status (Sox10+/Sox2+/Olig2- cells arrowheads, blood vessels asterisk, scale bar 20 µm). (E) SCs-mediated remyelination is restricted to the VN niche as shown by in situ hybridization for MPZ (I, scale bar 100 µm). Double ISH shows mutually exclusive and spatially separated pattern of *MPZ* and *PLP* transcripts (II, scale bar 20 µm). *MPZ* and *PLP* is clearly detectable as separable cell populations among Sox10+ cells in defined areas within remyelinated CNS lesions (III, scale bar 20 µm).

DOI: https://doi.org/10.7554/eLife.30325.002

The following figure supplements are available for figure 1:

**Figure supplement 1.** Characteristics of cells occupied vascular niche area in remyelination time course.

DOI: https://doi.org/10.7554/eLife.30325.003

**Figure supplement 2.** SCs-mediated remyelination is restricted to the vascular niche.

DOI: https://doi.org/10.7554/eLife.30325.004

(*Figure 1B,C*). The vast majority of these cells co-expressed the oligodendrocyte lineage transcription factor Sox10 (*Hornig et al., 2013*) and Sox2, a transcription factor associated with OPC activation and early stages of differentiation (*Zhao et al., 2015*). From 10 dpl there was a significant increase of the number of cells Sox10+/Sox2+ but that were Olig2- (*Figure 1C*). By 14 dpl up to 80% of total Sox10+ within the VN were Olig2- (av. 68.40 ± 14.65%, *Figure 1C,D*). These observations could have at least two explanations: 1) extensive apoptosis of Olig2+ OPCs takes place preferentially in vascular niche and is associated with proliferation of Olig2-/Sox10+ cells migrating to the lesion, or, alternatively 2) Sox10+ OPCs loose Olig2 expression within the VN. Since Sox10 is a transcription factor required for both OLs and SCs homeostasis and myelin maintenance (*Bremer et al., 2011*), we hypothesized that downregulation of Olig2 expression might be associated with switch in differentiation program and Olig2-/Sox10+/Sox2+ cells could give rise to cells producing peripheral-type myelin. To rule out cell death as the cause of the decrease in Olig2+ cells within the VN we performed immunohistochemistry for activated caspase three and found no significant differences in the number of apoptotic OPCs between the VN and non-vascular regions at any time points investigated (*Figure 1—figure supplement 1A*). Using Pdgfra-Cre reporter mice, we found that the Olig2-/Sox10+ cells at 14 dpl in the VN were likely of OPC origin (*Figure 1—figure supplement 1B*). These data suggest that oligodendrocyte lineage cells can lose Olig2 expression at early stages of differentiation while continuing to express Sox10 and Sox2, and that these cells, clustered within the VN are a source of SCs that eventually remyelinate the VN axons.

We also found substantial regeneration of blood vessels and BBB within the areas undergoing remyelination. Blood vessels rebuilding their basement membrane and endothelial walls occurred in parallel with remyelination (*Figure 1—figure supplement 1C*). Of note, perivascular SCs formed a network by contacting each other with no discontinuity of basement membrane of the wall of blood vessels and such capillaries rebuild BBB (*Figure 1—figure supplement 1F*).

To check if SC-mediated remyelination is confined to the VN we detected peripheral myelin proteins at 28 dpl by in situ hybridisation (ISH) for *myelin protein zero (MPZ)* mRNA and immunostaining for periaxin. Strong signal from *MPZ* mRNA only occurred in cells closely associated with the vasculature, suggesting that an interaction between progenitors and blood vessels might be necessary for their alternative, SC differentiation (*Figure 1E,I*). Double ISH revealed that transcripts for *MPZ* and *PLP* (encoding an oligodendrocyte specific protein) were mutually exclusive and spatially separated (*Figure 2E,II*). Moreover, *MPZ* and *PLP* segregation was also clearly detectable among cells expressed *Sox10* in defined areas within remyelinated CNS lesions (*Figure 2E,III*). Close examination of the VN revealed that periaxin+ SCs preferentially occupy discrete areas where astrocytes were absent even around the same vessel (*Figure 1—figure supplement 2*).

Our observations suggest that blood vessel itself or blood vessel-created microenvironment together with astrocyte absence are critical determinants for specific OPCs differentiation.

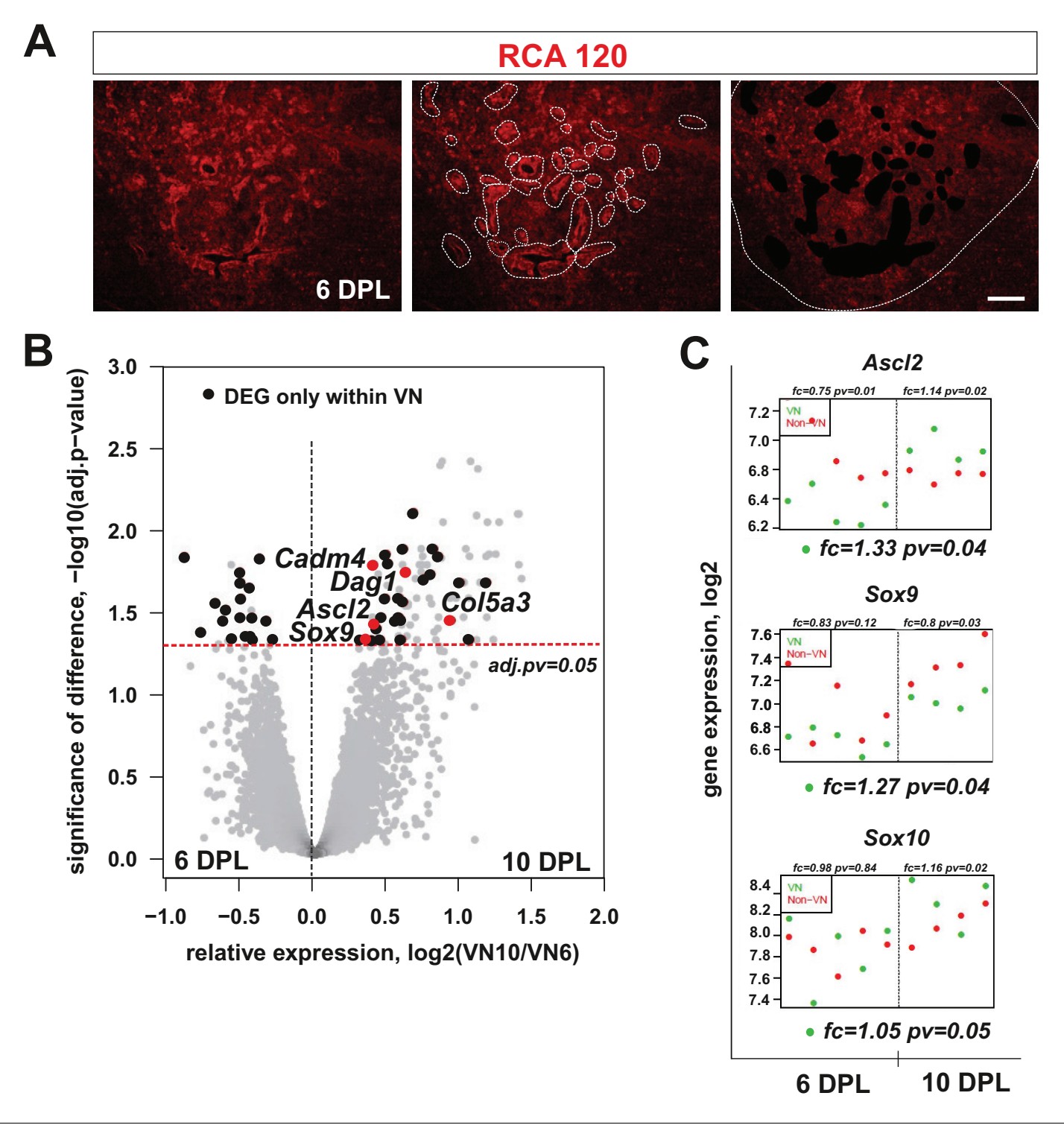

**Figure 2.** mRNA profiles differ in isolated microenvironmental niches. (**A**) Images illustrate the microdissection procedure: areas around blood vessels marked to be cut out by UV-laser (VN) and areas of lesion without blood vessels marked to be cut afterwards (non-VN) are depicted (scale bar 200 μm). (**B**) Volcano plot contrasting the significance [−log10(pv) on Y-axis] and the magnitude of the expression difference (log2 on X-axis) for comparison of gene expression between 6 and 10 dpl in VN. Each dot represents individual gene. Dashed line corresponds to the FDR acceptance level (adj. pv = 0.05, limma test, n = 4) (**C**) Transcript abundance level (absolute expression) of transcription factors in separated areas based on a hybridization intensity measurement for an individual probe, dots represent expression level for each animal sample, green – expression level in VN, red – expression

*Figure 2 continued on next page*

*Figure 2 continued*

level in non-VN. The significance of differences between niches at the given time is presented at the top of each diagram, separately for each niche. Fold changes between the time points 10 dpl vs 6 dpl in VN are shown at the bottom of each diagram.

DOI: https://doi.org/10.7554/eLife.30325.005

The following source data and figure supplement are available for figure 2:

**Source data 1.** Genes significantly regulated in the non-vascular areas.
DOI: https://doi.org/10.7554/eLife.30325.007
**Source data 2.** Genes significantly regulated in the vascular niche.
DOI: https://doi.org/10.7554/eLife.30325.008
**Source data 3.** Genes significantly regulated only in the vascular niche.
DOI: https://doi.org/10.7554/eLife.30325.009
**Figure supplement 1.** Microarray analysis restricted to defined and putative endothelial and astrocyte enriched transcripts shows the different profiles of both niches in 10 dpl group.
DOI: https://doi.org/10.7554/eLife.30325.006

## The demyelination-associated perivascular niche displays a specific pattern of gene expression

The differences in cellular composition between distinct lesion areas prompted us to investigate their transcriptomes. We hypothesised that they would reflect microenvironmental cues regulating OPC differentiation. To address this hypothesis, we first examined differences between VN and non-VN transcriptomes at 6 and 10 dpl, assuming that key determinants of alternative OPC fate should be activated at the time when early differentiation decisions are made. We used laser capture micro-dissection (LCM) to precisely separate pre-defined discrete tissue compartments. Since LCM procedure requires good visualization of a structure of interest we developed fast RAC120 staining in fresh tissue sections (*Figure 2A*) to define vascular and non-vascular areas. We collected tissue from vascular and non-vascular regions as separate pools from individual animal (n = 5 samples from both, VN and non-VN areas) and obtained high quality RNAs (with a RNA integrity number, RIN, above 7.0) that were used for global gene expression profiling using Illumina RatRef-12 microarrays.

Using bioinformatic tools we identified genes differentially expressed in separated niches. We found 115 genes with significantly changed expression in the VN between 6 and 10 dpl, and 313 genes with significantly changed expression in non-VN (adj.*p*-values<0.05; the source data are summarized in *Figure 2—source data 1–3*). Within genes upregulated in the VN we found several transcripts strongly associated with the endothelium. In contrast, the non-VN was enriched in several known astrocyte transcripts (*Figure 2—figure supplement 1*). We observed significant differences between the two area-specific transcriptomes only at 10 dpl, which suggested that the majority of molecular events determining OPC differentiation take place around this time.

Using differential gene expression (DEG) analysis, we identified a set of genes that were significantly regulated only in the VN and among them, several genes expressed exclusively in this microenvironment. A volcano plot, obtained by plotting the fold change of gene expression (log2 of 10 dpl/6 dpl ratio) against significance of the difference (-log10 adj.pv) revealed a prominent cluster of 18 such niche-specific genes at 6 dpl and 33 at 10 dpl (*Figure 3B*: black dots in the I and II quadrant of volcano plot, all data are shown in File S3). Among differentially expressed genes we found *Ascl2* and *Sox9*, coding for transcription factors previously implicated in SC differentiation programme (*Küry et al., 2002*; *Weider and Wegner, 2017*), significantly upregulated in the VN (*Figure 2C*, FC=1.33, FC=1.27, respectively, adj.pv<0.05). We also examined expression of *Sox10,* critically required for myelination in both CNS and PNS (*Hornig et al., 2013*; *Vogl et al., 2013*) and found it stably expressed within the VN (*Figure 2C*). In addition, several genes encoding for extracellular matrix proteins (ECM; *dystroglycan 1, Dag1; collagen type V alpha 3, Col5a3; cell adhesion molecule 4, Cadm4*) were significantly upregulated within the VN at 10 dpl. These data suggested that the distinctive VN transcriptome might be required for SC differentiation.

## BMP and Wnt ligands are prominent within the perivascular niche

To put the results of our differential expression analysis into a functional context we next focused on identifying transcriptional changes associated with complex signalling networks specific for the VN.

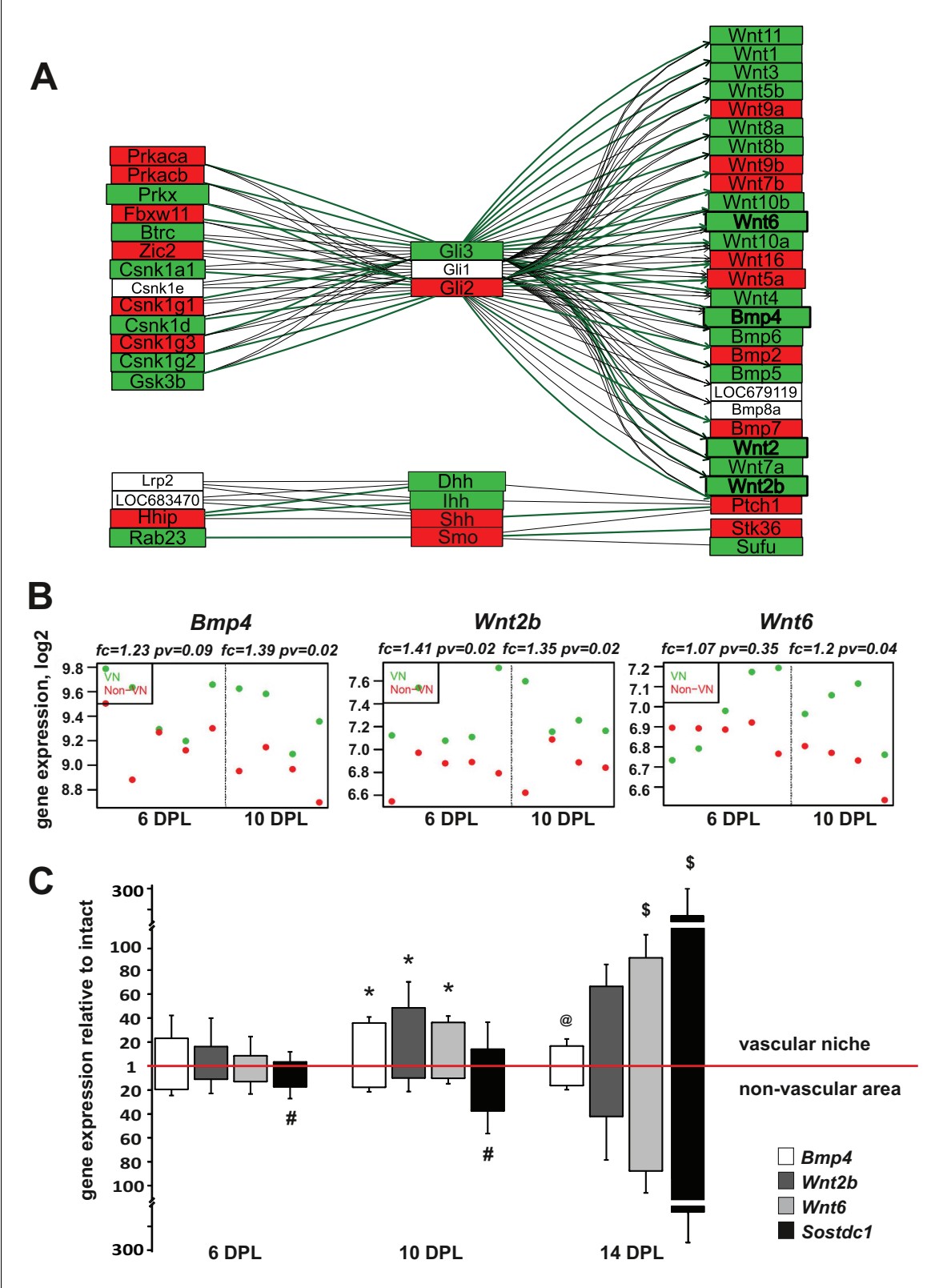

**Figure 3.** Injury-induced vascular niche is enriched with BMP and Wnt morphogen transcripts. (A) Graphical representation of BMP/WNT/Hedgehog signalling pathway interactions based on 10 dpl microarray results, higher expression of particular gene in non-VN in red, a higher expression in a VN group in green. Computed expression changes were marked over the existing topology using green-red colour scheme. Box size represents the relative difference of expression, genes with significant changes of expression bolded (Fisher's exact test, adj.p < 0.05). In order to highlight observed

*Figure 3 continued on next page*

*Figure 3 continued*

differential co-regulation of gene expression, consistent relations were marked with green. Two genes were assumed to be in consistent relation if changes of their expression levels between the compared groups were consistent with type of regulations between them (activation or inhibition). Such consistent relations indicate functionally related expression changes (*Mieczkowski et al., 2012*). (B) Absolute expression of *BMP4*, *Wnt2b*, and *Wnt6* in VN (green) and non-VN (red) at 6 and 10 dpl, dots represent expression level for each sample. (C) Relative gene expression (mean ± sd; n = 3, Mann-Whithney U-test, *p<0.001 compared with control levels of intact tissue as well as with non-VN at the same time point, # p<0.05 compared with control levels of intact, @p<0.01 compared with BMP4 level at 10 dpl, $p<0.001 compared with control levels of intact tissue as well as with 6 dpl and 10 dpl).

DOI: https://doi.org/10.7554/eLife.30325.010

The following figure supplement is available for figure 3:

**Figure supplement 1.** *BMP4,* but not *BMP2,* is upregulated within the demyelinated lesion, regardless of the specific area examined.

DOI: https://doi.org/10.7554/eLife.30325.011

To do this in an unbiased way, we downloaded definitions of 93 signalling pathways from KEGG database (2010). The definitions contained information not only about the presence of a given gene product, but also about interactions between the elements, co-regulation. Next, we placed the identified expression changes within pathways that contained significantly high number of differentially expressed genes. The most prominent group of genes clustered in the 'hedgehog signalling network', which includes interaction with elements of BMP and Wnt signalling pathways. Preserving the original pathway content and topology from KEGG database we generated a graphical representation of the expression data which visualizes relationship between genes expressed in VN and non-VN, respectively. It revealed the co-expression changes in the signalling network genes encoding for regulatory enzymes (protein kinase cAMP dependent α and β Prkaca, Prkaca; protein kinase X-linked, Prkx; ligases Fbxw11, Btrc; casein kinases Csnk; glycogen synthase kinase-3, Gsk3b; and interacting proteins Hhip, Rab23 to the left of the Gli transcription factors and effector proteins, mostly Wnt and BMP ligands to the right Separate group showed expression of hedgehog ligands Dhh, Ihh, Shh and their receptor Smo (*Figure 3A*). However, only genes coding for BMP4 and most of Wnt ligands were significantly upregulated in the VN. For the most discriminated genes the absolute changes in expression in both niches are shown in *Figure 3B* and *Figure 3—figure supplement 1*, for each animal separately.

We next validated of the microarray results by qPCR with mRNAs obtained from independent animals (n = 3). Employing the same technique for niche separation, we examined differences in expression level of the genes of interest between VN and non-VN (first normalised to the intact tissue). We extended our qPCR analysis to samples from 14 dpl to examine broader kinetics. qPCR results showed upregulation of all transcripts in both niches compared to the intact tissue and confirmed differences in expression of genes encoding for *BMP4* and *Wnt2b* between niches at different time points (*Figure 3C*). At 10 dpl significant upregulation of *BMP4* and *Wnt2b* expression was detected in the vascular niche compared to non-vascular niche.

BMP signalling is in part controlled by secreted antagonists. We found that Sostdc1 (sclerostin domain containing one protein) was the only BMP antagonist expressed within areas of demyelination; other BMP antagonists such as chordin, noggin, gremlin1, gremlin2, and follistatin were uniformly expressed at low levels in both compartments (FC of VN vs. non-VN for 6dpc and 10dpc, respectively: chordin: 0.96, 1.1, noggin: 0.94, 0.91, gremlin1: 1.05, 1.13, gremlin2: 0.99, 0.94, and follistatin: 0.95, 1, pv>0.05). At the time of highest *BMP4* expression, the expression of S*ostdc1* was significantly upregulated only in the non-VN (*Figure 3C*). *BMP*4 mRNA level decreased at 14 dpl in the VN, while the strong increase in *Sostdc1* expression occurred in both niches. Sostdc1 is a common BMP and Wnt signalling antagonist (*Laurikkala et al., 2003*; *Lintern et al., 2009*) and its absence from the VN at 6 and 10dpl would enable BMP and Wnt signalling to proceed un-antagonised. At 14 dpl, when Sostdc1 is highly expressed, equally in both compartments of the lesion, most of OPCs are already specified to either an oligodendrocyte or Schwann cell fate (*Woodruff and Franklin, 1999*; *Zawadzka et al., 2010*).

We next performed in situ hybridization using digoxigenin labelled cRNA probes to characterize expression, distribution and cellular sources of the identified morphogens. In the intact white matter, *BMP4* mRNA expression was below the limit of detection, but was increased after demyelination. The number of *BMP4*-expressing cells reached a maximum at 10 dpl, then decreased around 14 dpl (*Figure 4A*) and finally *BMP4* signal disappeared by 28 dpl (data not shown). *BMP4* expression was

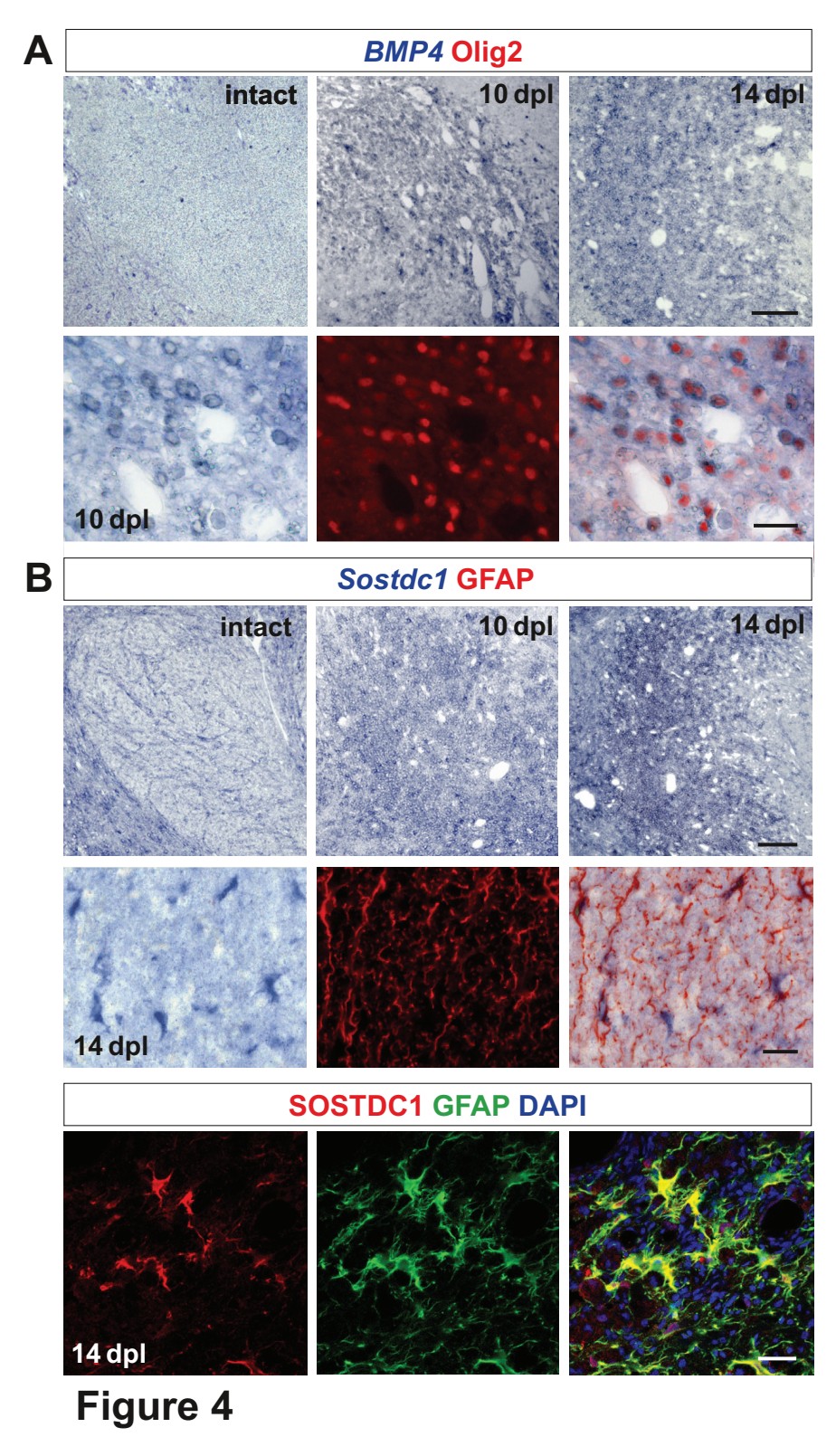

**Figure 4.** Spatiotemporal distribution and cellular sources of selected proteins. (A) *BMP4* transcript is produced in VN at 10 dpl (upper panel, in situ hybridization, scale bar 200 μm) mainly by OPCs (lower panel, Olig2 in red, scale bar 10 μm). (B) *Sostdc1* transcript detection by in situ hybridization and protein co-immunofluorescence show astrocyte association at 14 dpl (scale bar 200 μm upper panel, 20 μm middle and lower panels).

DOI: https://doi.org/10.7554/eLife.30325.012

*Figure 4 continued on next page*

*Figure 4 continued*

The following figure supplements are available for figure 4:

**Figure supplement 1.** Olig2-positive cells producing Sostdc1, likely pre-myelinating OLs, are located close to the lesion edge.

DOI: https://doi.org/10.7554/eLife.30325.013

**Figure supplement 2.** Gene expression analysis in the purified brain-derived cell populations at basal and activated conditions in vitro confirms the cellular origin of BMP4, Wnt2b, and Sostdc1.

DOI: https://doi.org/10.7554/eLife.30325.014

confined to oligodendrocyte lineage as demonstrated by immunolabelling of nuclei of *BMP4*-positive cells with Olig2. Since the temporal profile of Olig2+ cells closely resembled that observed for Olig2+/Sox2+ (*Figure 1E*), we identified *BMP4*+ cells as being injury-activated OPCs.

Transcripts for the BMP antagonist *Sostdc1* were not detectable in the VN at 6 and 10 dpl, while at 14 dpl it was expressed throughout the lesion area (*Figure 4B*). Co-localization of *Sostdc1* mRNA and GFAP at 14 dpl confirmed that astrocytes were its main source in the lesion (*Figure 4B*). However, some Olig2-positive cells, likely pre-myelinating OLs, also expressed *Sostdc1* (*Figure 4—figure supplement 1*).

Identification of cells expressing selected transcripts was explored in vitro under basal and wound healing conditions (*Figure 4—figure supplement 2*). Results of qPCR analysis confirmed that astrocytes (both, untreated and injured) were the main source of *Sostdc1*, while *BMP4* was produced by OPC and endothelium. The later also produced *Wnt2b* at a significantly higher level than any other cell types *in vitro*.

These data revealed spatiotemporal mutually exclusive *BMP4/Wnt2/Wnt2b* versus the BMP/Wnt antagonist *Sostdc1* expression within defined areas/niches of demyelinated white matter.

## Oligodendrocyte progenitor cells produce PNS myelin in a BMP and Wnt-dependent manner when transplanted into a PNS environment

We hypothesized that transplantation of OPCs to nervous tissue characterized by the absence of astrocytes, would mimic to some extent the environment of VN in demyelinated CNS white matter and thus uncover the capability of OPCs to terminally differentiate into functional SCs in vivo. First, we checked whether BMP and Wnt ligands as well as Sostdc1 were expressed in injured peripheral nerve. We found that *Bmp4* and *Wnt2* transcripts but not *Sostdc1* were significantly increased early after sciatic nerve injury (*Figure 5A*).

As the pattern of gene expression resembled that in CNS vascular niche and denuded axons were temporally present, we injected $1.6 \times 10^5$ GFP+ OPCs isolated from rat primary cultures into the sciatic nerves of adult rats at 24 hr after injury. Some rats received OPCs pre-treated with recombinant human USAG1 or dorsomorphin to interfere with BMP/Wnt and BMP signalling activation, respectively. Additionally, untreated cells were also grafted into sciatic nerves at 3 and 6 dpc to examine their fate in different conditions at the time of implantation.

At 28 days after crush we found GFP+ cells in longitudinal sciatic nerve sections in all experimental groups (*Figure 5B*): however, their number, morphology and antigenic characteristics differed substantially. GFP+ cells implanted at 24 hr after injury were broadly distributed around the injection place and displayed distinctive morphology typical for PNS-type SCs, positive for periaxin, a protein occurring only in PNS-myelin, formed a 1:1 association with axons, and clear organization of internodes (74.7 ± 10.25% of GFP+ cells, *Figure 5B,C*). We found also GFP+ cells that did not contact axons and remained periaxin negative (*Figure 5C*). The cells pre-treated and injected in the presence of dorsomorphin were present in recipient nerves, but they rarely produced peripheral myelin protein and did not formed the myelin sheaths (*Figure 5B,C*). Many patches of GFP+ cells were found also in sciatic nerves injected with USAG-pretreted OPCs: however, they were present close to the injection site, formed an irregular mesh, rarely contacted with an individual axon to form myelin sheath and produced periaxin (*Figure 5B,C* and *Figure 5—figure supplement 1*).

In contrast to OPCs grafted into sciatic nerves at 24 hr after crush, untreated cells implanted at three dpc did not produce periaxin after 28 dpc. Also, they did not produce GFAP, normally expressed by astrocytes in CNS and non-myelinating SCs in the PNS as well as they lost the expression of Olig2, constitutively expressed by oligodendrocyte lineage cells (*Figure 5—figure*

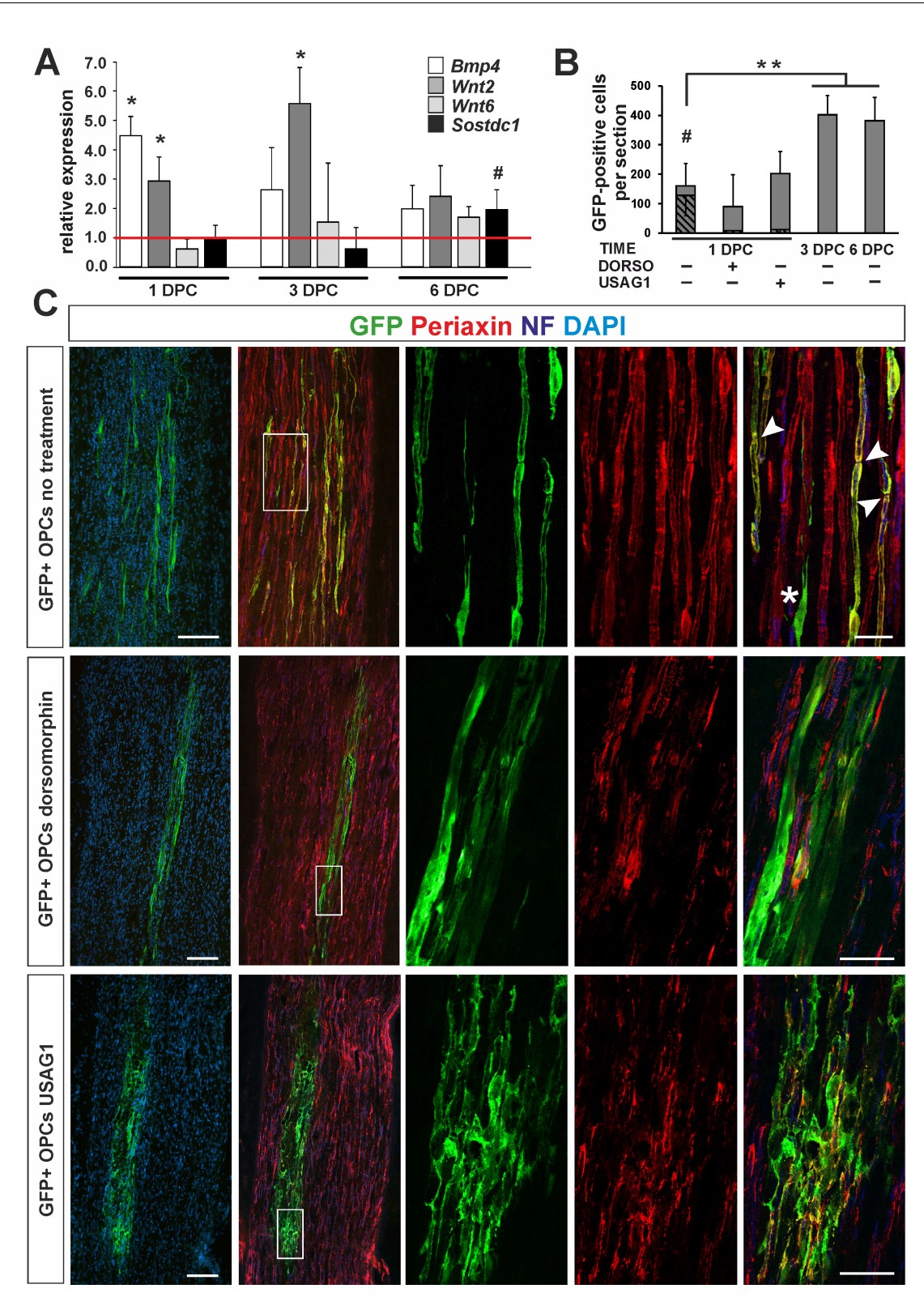

**Figure 5.** OPC differentiation fate at 4 weeks after transplantation into injured sciatic nerve. (**A**) Gene expression pattern after sciatic nerve crush (mean ± sd; n = 3, Mann-Whitney U-test, *p<0.01 compared with 30 min after crush, #p<0.05 compared with 3 dpc). (**B**) Quantification of GFP+ cell number in longitudinal sciatic nerve section in different transplantation paradigms (mean ± sd, three fields per section, three sections per nerve, n = 3 rats per experimental condition, t-Students test, **p<0.01, quantification of GFP+/periaxin+ cells shows their significant contribution in nerves were

*Figure 5 continued on next page*

*Figure 5 continued*

cells were implanted at 1dpc, shaded bars, # p<0.01). (**C**) Immunostaining shows grafted cells existing within sciatic nerves at each experimental condition. The first two images of each panel show general view of longitudinal sections in low power, the next three show boxed area reimaged using confocal microscope. Co-localization of GFP (in green), with periaxin (in red) indicates SC differentiation of OPCs grafted into sciatic nerves at 24 hr postinjury, (arrowheads). Non-myelinating GFP+ cells could be also detected (asterisk). SCs differentiation is defective under both dorsomorphin and USAG1 treatment (scale bar 100 μm and 20 μm in low and high power images, respectively).

DOI: https://doi.org/10.7554/eLife.30325.015

The following figure supplement is available for figure 5:

**Figure supplement 1.** OPCs differentiate abnormally when grafted into sciatic nerve at 3 or 6 dpc.

DOI: https://doi.org/10.7554/eLife.30325.016

*supplement 1*). OPCs were highly proliferative when transplanted to the sciatic nerve at 3 and 6 dpc and the cells populated large areas of the nerve (*Figure 5B* and *Figure 5—figure supplement 1*).

To verify the causative role of Wnt signalling and BMP/Wnt cooperation in OPC fate modulation, we infected OPCs with lentiviral shRNA against Lrp6, a Wnt co-receptor necessary for the signaling pathway induction, mimicking Sostdc1 action with regard specifically to Wnt and examined their ability to differentiation in vivo. 61 ± 4.2% and 56.4 ± 1.2% of GFP+ cells were transduced with LRP6 shRNA and control, non-targeting shRNA, respectively (*Figure 6A*). Next, we FACS-sorted GFP/tRFP double positive cells after 3 div postinfection using highly restricted conditions for sorting purity (*Figure 6B*). We determined efficacy of gene expression silencing at the protein level by densitometry of western blot signals in OPCs as 73 ± 4,3% compared to non-targeting control shRNA-transduced cells (mean ±SEM of two experiments, *Figure 6C*). No significant differences in LRP6 protein level were observed between the untreated control and non-targeting shRNA-transduced cells. Transduced cells differentiated normally into MBP-produced oligodendrocytes at 6 div in differentiation medium (*Figure 6D*).

To test differentiation fate of OPCs with Wnt co-receptor knockdown we implanted FACS-sorted cells into sciatic nerves at 24 hr after crush injury. As shown in *Figure 6E,F* we found GFP/tRFP double positive cells in longitudinal host sciatic nerve sections after 28 dpc. However, only control non-targeting shRNA-transduced GFP+/tRFP+ cells showed typical morphology of MPZ-producing Schwann cells within host sciatic nerves. For quantification we detected intracellular Schwann cell protein, periaxin and demonstrated that Schwann cell differentiation of LRP6 shRNA lentiviral transduced OPCs was significantly impaired; only 15.8 ± 12% of GFP+ cells were periaxin-positive in contrast to the control shRNA-transduced cells (73.6 ± 16.9% differentiated into Schwann cells).

Thus, we found that pharmacological or genetic inhibition of Wnt signaling inhibited Schwann cell differentiation of OPC within injured sciatic nerve suggesting a key role of Wnt in cell differentiation decision.

These results suggest that activation of both BMP and Wnt signalling might be required for initiation of OPC alternative differentiation.

## CNS-derived SCs do not contribute to CTGF production within remyelinated lesion

The extracellular matrix proteins were the second functional group of genes found to be specifically enriched in the VN and the most substantially upregulated being genes encoding for collagens (*Col5a3, Col11a1, Col1a2*) (*Figure 7A*). In contrast, within non-VN there was increased expression of vimentin (*Vtn*) and fibronectin (*Fn1*). A particularly highly expressed gene within the non-VN was connective tissue growth factor (*CTGF*). CTGF, a member of the CCN family of secreted matricellular proteins, can bind multiple ligands and interact with many other proteins in the remodeling of lesion environments. Its interactions with the ECM and integrins have been reported to affect oligodendrocyte differentiation (*Colognato et al., 2002*). Moreover, CTGF has been recently identified as a SC-derived factor inhibitory for myelination (*Lamond and Barnett, 2013*). As shown by microarray data and qPCR, *CTGF* expression decreased gradually during CNS remyelination, with expression levels consistently lower in the VN than in non-VN (*Figure 7A and B*). The lowest expression of *CTGF* was detected in VN at 14 dpl, when the first mature SCs were already present in the niche.

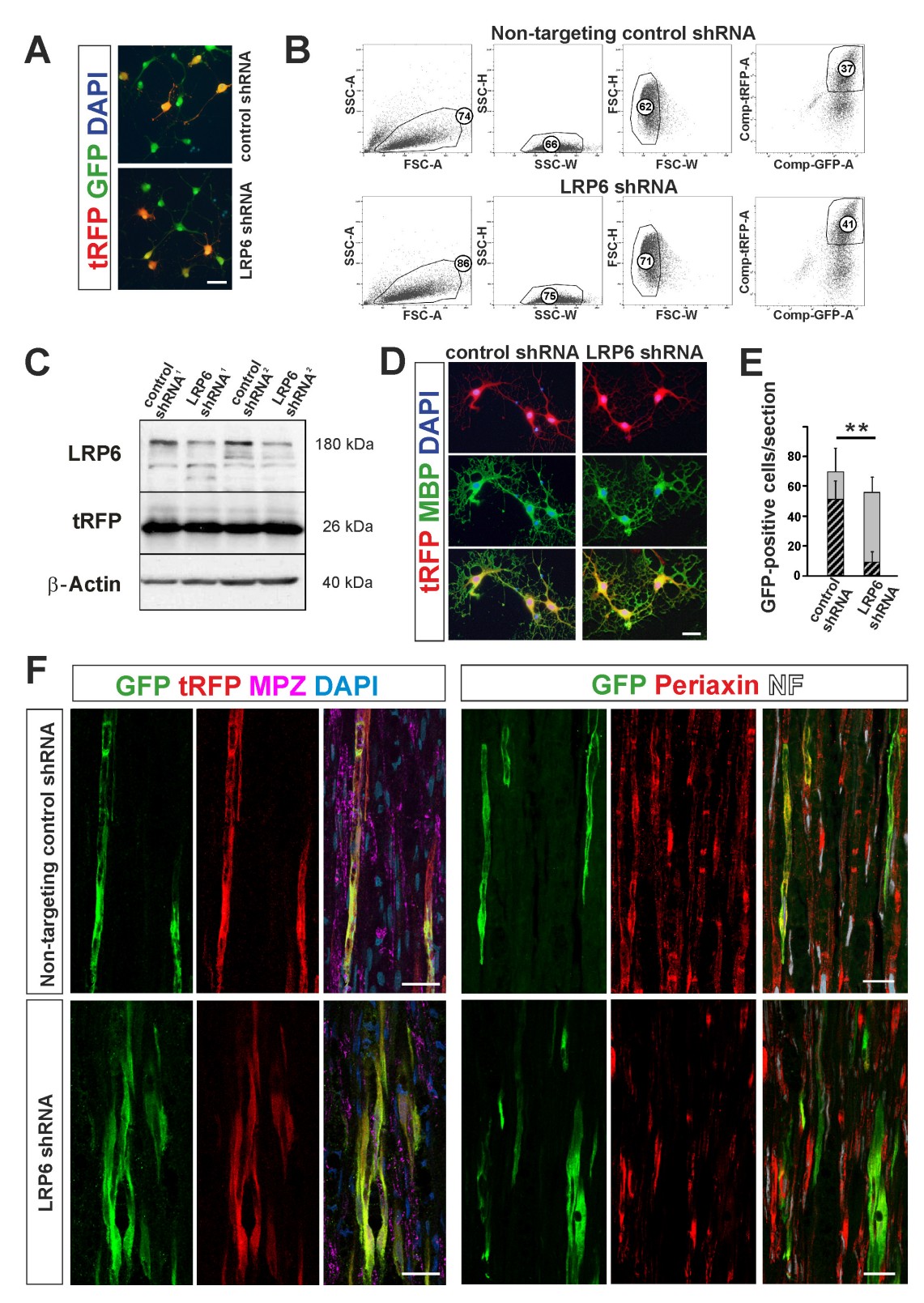

**Figure 6.** shRNA-mediated knockdown of LRP6 expression inhibits Schwann cell differentiation within injured sciatic nerve. (**A**) Images show morphology of GFP+ OPCs transduced in vitro with lentiviral-mediated control and LRP6-specific shRNA, scale bar 20 μm (**B**) FACS sorting strategy, numbers show percent of cells of a specified phenotype (**C**) Western blot analysis of LRP6 and tRFP protein level revealed efficient gene expression knockdown. (**D**) FACS-sorted GFP+ OPCs transduced with LRP6shRNA differentiate normally into MBP+ OLs, scale bar 20 μm (**E**) Differentiation of

*Figure 6 continued on next page*

*Figure 6 continued*

OPCs transduced with LRP6-specific shRNA into SCs is defective. t-Students test **p<0.01 (**F**) Confocal images show grafted GFP+/tRFP+ cells differentiated within host sciatic nerves at 4 weeks after implantation. Co-localization of GFP (in green), with MPZ or periaxin (in red) indicates SC differentiation of OPCs transduced with non-targeting control shRNA but not with LRP6 shRNA (scale bar 20 μm).
DOI: https://doi.org/10.7554/eLife.30325.017

To address whether ectopic SC differentiation contributed to CTGF production as occurs in the PNS, we specifically labelled peripheral glia in transgenic mice that allowed us to compare the level of CTGF production in CNS-derived and PNS-derived cells, separately.

We used mice carrying a Cre recombinase under the control of Wnt1 promoter, that is normally active in embryonic neural crest precursors, and a YFP reporter gene with an upstream floxed stop cassette in the Rosa26 locus (Wnt1-Cre;R26-YFP mice) in which all neural crest progeny were labelled. We first demonstrated that Wnt1-Cre was active in SCs in peripheral nerves, but not in OPCs (*Figure 7C*). We then induced focal demyelination to the Wnt1-Cre mice and found that very few MPZ+/YFP+ SCs were detected within the lesions at 28 dpl, even if the lesion was located next to the ventral root and despite the high efficiency of labelling of peripheral cells. MPZ+ myelin within the CNS lesion was almost exclusively associated with cells negative for YFP (*Figure 7C*). In striking contrast to peripheral SCs, cells that remyelinated spinal cord white matter did not produce CTGF (*Figure 7D*). The CTGF+ cells were mainly detected at the lesion rim and displayed the typical stellate morphology of activated astrocytes (*Figure 7D*). These findings strongly suggest that expression and accumulation of CTGF is restricted to the astrocytic response to demyelination. Furthermore, we demonstrate that mature CNS-derived SCs differ from their PNS counterparts that supports the previously postulated similarity between them and OLs rather than *bona fide* PNS glia (*Imbschweiler et al., 2012*).

## Discussion

In this study, we demonstrate that CNS-derived SCs selectively occupied the perivascular area devoid of astrocytes and propose that differentiation of OPCs into SCs depends, at least partially, on vascular reorganization.

We show that the tissue around the blood vessels promotes preferential accumulation of activated OPCs early after demyelinating insult, which correlates with remodelling of the capillary network. This is in agreement with data reporting that cerebral endothelium (*Hayakawa et al., 2011*; *Tsai et al., 2016*) promote OPC migration along blood vessels during development. These two processes are likely interconnected, indicating that OPC activation and their paracrine activity integrates the onset of myelination with white matter angiogenesis also in adulthood.

Identification of a perivascular niche as a causative factor for alternative differentiation prompted us to employ transcriptome profiling approach in order to describe signals and processes that occur in the niche. Comparative analysis of global transcriptome in discrete areas of the same affected structure pointed towards BMP and Wnt signalling pathways as the crucial players with context-dependent functions. The implication of BMPs in healthy brain neurogenesis (*Colak et al., 2008*) and after remyelination has been demonstrated. BMPs interfere with OPCs differentiation after toxin-induced demyelination, whereas BMP antagonists such as noggin promote OL generation and remyelination (*Sabo et al., 2011*). The infusion of noggin or chordin increases the commitment of SVZ cells to the oligodendrocyte lineage and their mobilization following demyelination (*Jablonska et al., 2010*; *Cate et al., 2010*). Furthermore, BMP signalling interferes with OL maturation following perinatal hypoxic-ischemic brain injury (*Dizon et al., 2011*) or intraventricular hemorrhage (*Dummula et al., 2011*). BMP4 affects myelination by downregulating the transcription factors, Olig1 and Olig2, which are required for OL differentiation (*Samanta and Kessler, 2004*; *Cheng et al., 2007*). Altogether, these data indicate that BMP acts as a negative regulator of OL differentiation by preventing cell mobilization and maturation and promotion of astrogliosis.

We identified BMP4 as an OPCs-derived factor whose increased expression in the VN did not, however, correlate with astrogenesis. Moreover, as we showed previously, adult OPCs do not differentiate significantly into astrocytes in vivo, only occasionally we found OPC-derived astrocytes in the

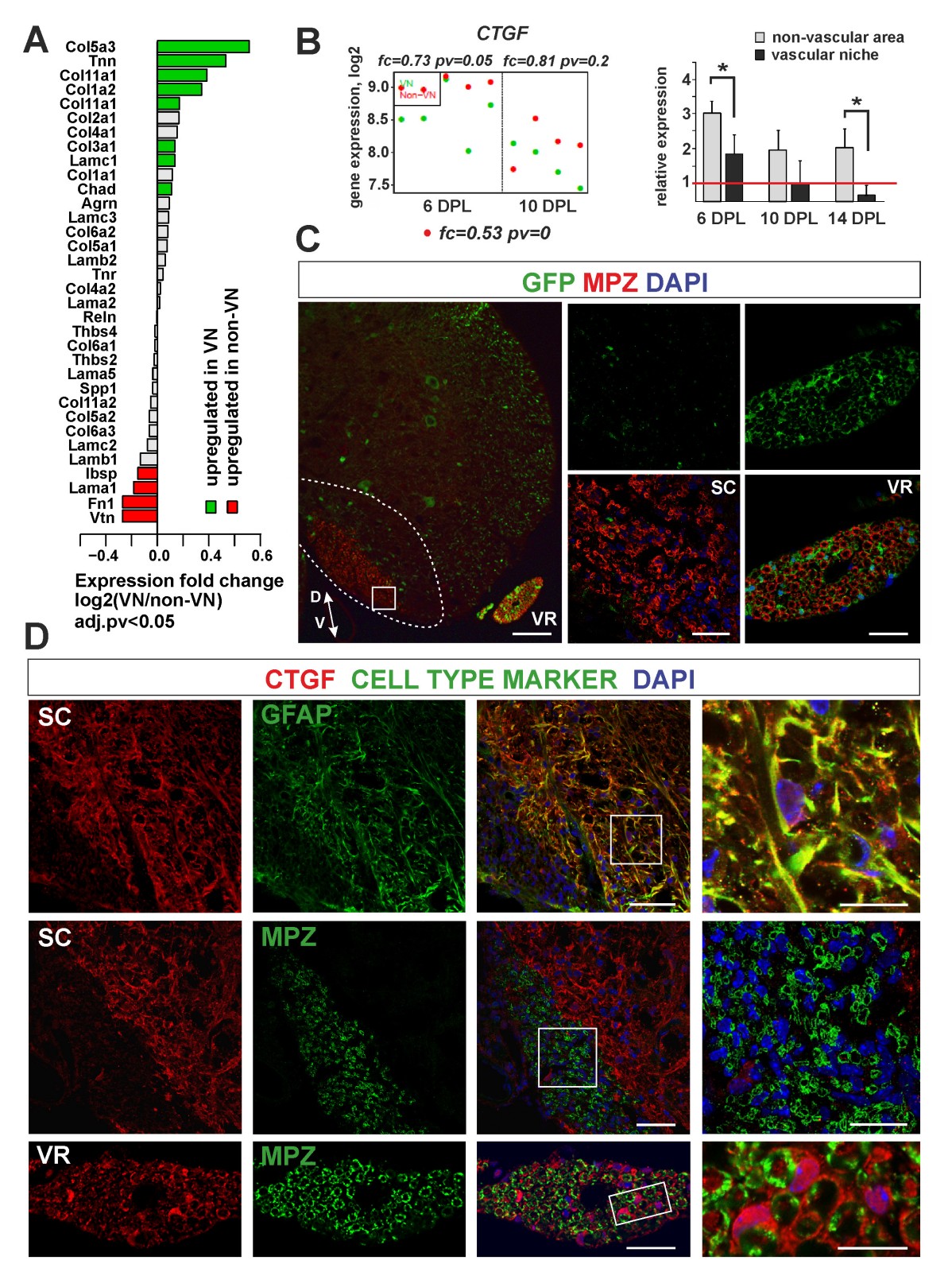

**Figure 7.** Expression and accumulation of CTGF is restricted to the astrocytic response to demyelination. (**A**) Expression of ECM proteins is niche-specific as revealed by microarray analysis (Fisher's exact test, adj.pv <0.05). (**B**) *CTGF* expression decreases during remyelination; however it is significantly higher in non-VN than VN (microarray data on the left, relative expression over intact on the right, t-Students test **p<0.01). (**C**) Representative images of typical lesion epicentre in the ventral spinal cord white matter of Wnt1-Cre mice at 28 dpl. MPZ associated with YFP-negative

*Figure 7 continued on next page*

*Figure 7 continued*

cells within CNS lesion indicate their other than neural crest origin in contrast to *bona fide* PNS SCs myelinating axons in the ventral root (vr). High confocal power images (to the right) show merged immunostaining of boxed area in spinal cord lesion (sc) and ventral root (vr). Scale bar 200 μm and 50 μm in low and high power images, respectively. D, V dorso-ventral position of the lesion. Lesion depicted by dotted line. (D) Co-localization of CTGF with GFAP in remyelinated spinal cord white matter or with MPZ in ventral root (scale bar 50 μm, overlaid image of boxed areas, upper 15 μm, middle and lower 20 μm).

DOI: https://doi.org/10.7554/eLife.30325.018

non-vascular areas (*Zawadzka et al., 2010*). Our results suggest that BMP4 takes part in the autonomous, alternative PNS-like, OPC differentiation.

BMP4 has been proposed to promote differentiation of engrafted adult OPCs primarily into cells with immunophenotypic and ultrastructural characteristics of myelinating SCs (*Talbott et al., 2006*). SC differentiation was blocked when OPCs were co-transplanted with astrocytes or by noggin overexpression in engrafted OPCs. However, neither noggin overexpression nor the presence of co-transplanted astrocytes enhanced differentiation into OLs, suggesting that suppression of BMP signalling alone is not sufficient to promote OL differentiation. Since endogenous OL remyelination is normally successful in the perimeter of lesions, coinciding with a region of densely populated reactive astrocytes (*Blakemore and Crang, 1989*; *Franklin and Blakemore, 1993*; *Talbott et al., 2006*) it was unclear why co-transplantation of astrocytes with adult OPCs did not induce OL differentiation. Our results suggest that BMP4 is necessary but not sufficient to induce PNS-like differentiation program. Upon demyelination OPCs upregulate BMP4 and loose Sosctdc1 expression which enable them to convert into activated cells, primed for differentiation. This corroborates previous studies on OPC plasticity. In particular, it has been shown that BMP4 treatment inhibits the progression of OPC differentiation toward OLs by favouring global histone acetylation and nuclear chromatin conformation characteristic for euchromatin. Thus, BMP4 acts as a potent inducer of gene expression, possibly supporting the undifferentiated progenitor state and/or making the cells more responsive to the other signals such as Wnt. In contrast, OL differentiation is accompanied by peripheral chromatin condensation in OPCs (*Wu et al., 2012*) and decrease in histone deacetylation at transcriptional repressors of oligodendrocytic differentiation such as Sox11 (*He et al., 2007*) and Sox2 (*Lyssiotis et al., 2007*). We show that BMP4 expression coincides with OPCs activation and this provides an additional aspect to the notion that OPCs can acquire multipotency in specific conditions of local microenvironment. Consequently, astrocyte-derived Sostdc1 controls biological availability of BMP4 in the non-vascular area and maintains OL-driven remyelination. Since Sostdc1 is produced exclusively by activated astrocytes within the non-VN lesion area during the period of first 10 days post demyelination our data corroborate with the long-postulated notion that astrocytes are one of the key regulators of OPC differentiation fate.

Our in vivo experiments provide evidence for the ability of OPCs to generate SCs cells in the PNS under the condition resembling the CNS perivascular niche, in a BMP and Wnt dependent manner. OPC differentiation was affected by both USAG1 and dorsomorphin treatment. USAG1 (a product of human ortholog of rat gene *sostdc1*) binds directly to BMPs and antagonizes BMP signalling (*Yanagita et al., 2006*) and interacts with the LRP6 receptor to antagonize Wnt signalling (*Lintern et al., 2009*). Dorsomorphin is a potent inhibitor of BMP signalling which binds to the BMP type I receptors ALK1/2/3/6 and type II receptor ActRIIA (*Horbelt et al., 2015*). Thus, BMP stimulation seems to be indispensable for the alternative fate induction. Moreover, inhibition of Lrp6-dependent Wnt signaling prevents OPC-to-SC differentiation even within the high BMP4 background of injured sciatic nerve. Defective phenotype of cells transduced with LRP6 shRNA indicates that activation of WNT signalling is required for initiation of PNS fate differentiation which has not been proposed yet.

Our findings demonstrate that OPCs differentiate into SCs when exposed to BMP/Wnt-enriched PNS environment; however, an axonal contact seems to be an indispensable factor. This is consistent with recent findings showing that neuregulin-1 (Nrg1) mediates SCs remyelination by precursors of CNS origin after spinal cord injury (*Bartus et al., 2016*). However, while neuregulin-1 is evidently important for Schwann cell myelination of the CNS axons, we did not find any evidence in our whole genome analysis to suggest that Ngr1 is involved in OPCs fate determination at its early stage.

Finally, by using a transgenic approach we have provided novel evidence that CNS-derived SCs substantially differ from PNS myelinating cells, that is by different CTGF production. We show that CTGF participates in the pathological reaction of the astrocytic compartment which is in line with previous reports describing contribution of accumulating CTGF+ astrocytes to wound healing and glial scar formation in response traumatic brain injury (*Hertel et al., 2000*).

This study is the first to demonstrate the PNS-like OPC differentiation effect of BMP and Wnt signalling, both of which are signalling pathways associated with inhibition of OL differentiation and maturation. We propose that BMPs induce OPC priming and activation and, when combined with Wnt, instruct them to differentiate into peripheral myelin producing cells. This sets the stage for further investigations into the downstream effectors of the proposed signalling networks that govern alternative OPC differentiation during CNS remyelination.

## Materials and methods

### Animal studies

All research and animal care procedures were approved by the First Warsaw Local Ethics Committee for Animal Experimentation (protocols 1020/2009, 1046/2009, 556/2014 and 431/2017) and performed according to international guidelines on the use of laboratory animals.

The animals were kept under standard laboratory conditions on a 12 hr light/dark cycle with constant access to food and water and were randomly assigned to experimental groups.

Animals used:

8–10 week-old female Wistar rats;

9–12 week-old female Wnt1-Cre transgenic mice expressing the Cre recombinase under control of the Wnt1 promoter elements described previously (*Danielian et al., 1998*). Homozygous or heterozygous Wnt1-Cre mice were crossed with homozygous Rosa26-YFP reporters to generate double-heterozygous offspring for experiments. In Wnt1-Cre double-heterozygous mice, virtually all of SCs in the ventral and dorsal roots became YFP-labelled (since the promoter is active at premigratory neural crest cells, *ibidem*);

9–12 week-old female PDGFRα-CreERT2 transgenic mice expressing the Cre recombinase under control of the PDGFRα promoter described previously (*Rivers et al., 2008*). Homozygous or heterozygous PDGFRα-CreERT2 mice were crossed with homozygous Rosa26-YFP reporters to generate double-heterozygous offspring for experiments;

P0-P2 pubs of transgenic eGFP-expressing rats Wistar-TgN(CAG-GFP)184Ys, National Bio Resource Project for the Rat in Japan, Kyoto University.

### Cell lines

Rat brain microvascular endothelial cells were purchased from Innoprot. Primary rat astrocytes and OPCs were isolated as described in the method details section, In vitro cell culture and treatment. Cells were maintained at 37°C in a humidified atmosphere of 5% $CO_2$.

### Surgical procedures

Demyelination in rats was induced bilaterally by stereotaxic injection of 4 µL of 0.01% ethidium bromide (EB, vol/vol, Sigma) into the caudal cerebellar peduncles (CCP, 10.6 mm caudal,±2.6 mm lateral and 7.9 mm ventral to bregma) as previously described (*Woodruff and Franklin, 1999*). EB was delivered at a rate of approximately 1 µL per minute and the injection needle remained in position for 4 min.

In mice, Cre recombination was induced by administering tamoxifen (40 mg/ml, Sigma), dissolved in corn oil by sonication for 45 min at 30°C. Adult mice were given 300 mg per kg of body weight by oral gavage or by intraperitoneal injection, at 100 mg per kg of body weight, daily for four consecutive days. Tamoxifen treatment was stopped 5 days prior to inducing demyelination.

Demyelination in mice was induced by injection of 1 µL of 0.1% ethidium bromide (EB) into the ventral or dorsal white matter funiculus at the level of T12 as previously described in detail (*Zawadzka et al., 2010*; *Zhao et al., 2015*). The position of T13 was identified and the epaxial musculature was cleared from the immediate area. The space between T12 and T13 was exposed and carefully cleared, the central vein was identified, and the dura was pierced with a dental needle

lateral to the vein. A three-way manipulator was then used to position the needle for stereotaxic injection of EB. Hamilton needle with a fine glass tip was advanced through the pierced dura at an angle appropriate for ventrolateral or dorsal funiculus injection. Injection was controlled at 1 μL per minute and the needle remained in the injection site for 2 min to allow maximal diffusion of toxin.

## Controls received whole surgical procedure without toxin injection.

For sciatic nerve crush in rats an incision was made over the length of the right hip and, after exposing the nerve, constant pressure was maintained for 30 s with using watchmaker's forceps. For intra-neural cell transplantation rats were randomly divided into groups receiving: 1. OPCs in control medium (0.1% BSA (wt/vol), 0.1% DMSO in DMEM), 2. OPCs pretreated with a recombinant human USAG1 protein (human *Sostdc1* gene expression product) (2 μg/mL, R and D Systems, carrier-free, in DMEM), and 3. OPCs pretreated with dorsomorphin (5 μM, Sigma). The sciatic nerve was re-exposed at the indicated time at the level of the previous injury and injected with 8 μL of cell suspension in density of $2 \times 10^4$ cells per microliter into four sites along the nerve over 4 mm proximal-distal to the lesion epicentre using a Hamilton syringe with a glass micropipette attached on the top, at a rate of 1.0 μL/min. Lentiviral-transduced cells were injected in the same manner but animals received cell suspension in density of $10^4$ cells/μL to avoid cell clustering. The procedure was performed under a surgical microscope to visually confirm swelling of the nerve during intraneural injection. The animals in which the injected solution escaped from the needle track during injection or after withdrawal of the needle were omitted from the experiment. During all surgical procedures animals were anaesthetized with isoflurane supplemented with buprenorphine (0.03 mg/kg, intraperitoneal injection) for pain relief.

## In vitro cell culture and treatment

Primary mixed glial cultures were prepared from P0-P2 Wistar rat pups. Briefly, meninges were removed from brain hemispheres, forebrains were enzymatically (0.025% trypsin at 37°C for 20 min) and mechanically dissociated to a single cell suspension and cells were plated at a density of $3 \times 10^5$ cells/cm$^2$ on poly-L-lysine-coated flasks in Dulbecco's modified Eagle medium (with Glutamax, 4.5 g/L glucose, 10% heat-inactivated foetal bovine serum (FBS, Gibco), 100 U/mL penicillin, and 0.1 mg/mL streptomycin). After 8 days, microglial cells were removed by mild shaking over 1 hr and then cultures were shaken for additional 16 hr on a rotary shaker at 37°C at 250 rpm. OPCs were collected as the floating fraction. Adherent cells, astrocytes, were re-seeded on poly-L-lysine-coated culture plates at a concentration of 50 cells/mm$^2$ and maintained for 48 hr before treatment in DMEM containing 10% fetal calf serum (with 2 mM glutamine and antibiotics).

Astrocytes and endothelial cells were activated to proliferation and migration in wound healing conditions. Cells were plated on a 35 mm Petri dish, and the cell monolayer was scraped off the in five straight parallel lines using a 100 μl pipette tip to create 'scratches'. After washing with phosphate-buffered saline (PBS) cells were maintained in standard medium for indicated time.

For grafting into sciatic nerve GFP+ OPCs were isolated from primary glial cultures obtained from W-Tg(CAG-GFP)184Ys rats as described above and floating cells fraction were subsequently incubated with the tested compounds for 1 hr on rotary shaker at 100 rpm. Afterwards, the cells were carefully pelleted and re-suspended at the density of $2 \times 10^4$ cells/μL in the same medium containing respective compounds for implantation.

For viral transduction GFP+ OPCs were seeded at a density of $2 \times 10^6$ per poly-l-lysine coated T-75 cm$^2$ flask in OPC medium (DMEM supplemented with Sato's medium, 10 ng/ml PDGF-AA and 10 ng/ml bFGF (both form PeproTech). Cells were infected with a set of three on-TARGET plus SMARTvector Lentiviral shRNAs for rat LRP6 (shRNA-LRP6, targets ORF) and non-targeting shRNA (shRNA-control) synthesized by Dharmacon. All constructs were driven by constitutive cytomegalovirus (CMV) promoter and expressed TurboRFP fluorescent reporter. The sites targeted by the LRP6 siRNA lentivirus were: ATTATAACAAAGCGACTTG, ATGGGATCTAACACAATAG, and GGGTAATC TAAGCCGAACT, the titer was $10^8$ TU/mL. For transduction, OPC were exposed to lentiviral particles at a multiplicity of infection (MOI) of 2. The cells were incubated in viral supernatants for 8 hr, and then the culture medium was replaced. At 72 hr the transduction efficiency was assessed as the ratio of the number of tRFP-positive cells to the total GFP-positive cells.

## FACS sorting

After 72 hr post infection OPCs were dissociated with papain solution (STEMCELL Technologies), blocked with 20% FCS in DMEM and subsequently suspended in ice-cold PBS. tRFP-positive cells were sorted out of GFP-positive OPC with FACS Aria II sorter using FACS Diva software (BD Biosciences). Cell aggregates were removed from analysis during each experiment by doublet exclusion using pulse processing-based FSC-H vs FSC-W method. Following sorting double positive cells were implanted into sciatic nerves at 24 hr after crush injury as described above. A portion of cells was plated at the density of $6 \times 10^4$ per 24-wells plate and switched to differentiation in OPC medium without growth factors. Cells were fixed with 4% paraformaldehyde before being analysed for differentiation by immunostaining for myelin basic protein. Efficacy of gene knockdown was determined by western blot analysis.

## Protein isolation, electrophoresis and detection

Whole-cell protein extracts were separated on SDS-PAGE before electrophoretic transfer onto a nitrocellulose membrane (Amersham Biosciences). The membranes were incubated with primary antibodies diluted in a blocking buffer overnight and then with relevant secondary antibodies for one hour. Antibodies recognizing LRP6 (diluted 1:1000) was purchased from Abcam, tRFP (diluted 1:2000) from Evrogen, β-Actin antibody (diluted 1:2000) from Oncogene Research Products, horseradish peroxidase-conjugated anti-rabbit IgG (diluted 1:2000) from Vector Laboratories. Immunocomplexes were visualized by using ECL (Amersham). The molecular weight of proteins was estimated with pre-stained protein markers (ThermoScientific). Band intensities were determined by densitometric analysis of immunoblots with Molecular Imager FX and Quantity One software (BioRad).

## Tissue processing

For fresh-frozen tissue isolation animals were terminally anaesthetized with pentobarbitone at appropriate time after demyelination (n = 5 for each time point), brains were immediately removed and frozen in isopentan alcohol. Brains were cryosectioned into 20 µm slices and mounted on PEN membrane (polyethylene naphthalate, Arcturus, ThermoFisher Scientific) for further laser captured microdissection. For in situ hybridization and immunodetection animals were terminally anaesthetized with pentobarbitone and intracardially perfused with 4% (w/v) paraformaldehyde (PFA, Sigma) in phosphate-buffered saline (PBS, pH 7.4) at the indicated time after surgical procedure. Lesion containing tissue was dissected, post-fixed in 4% PFA overnight then immersed in 30% sucrose solution prepared with PBS for 48 hr before embedding with OCT (ThermoFisher Scientific). 12 µm coronal or longitudinal sections were thaw-mounted onto Poly-L-lysine-coated slides and stored at −80°C until further use.

## Global gene expression analysis

Brain vessels was visualized in tissue sections by incubation with rhodamine labelled *Ricinus Communis* Agglutinin I (RCA120, 1:500 in PBS, Vector Laboratories) on ice for 5 mins and immediately dehydrated. All solutions were prepared using RNAse free water. Predefined areas within lesioned CCP were dissected form each section through the whole lesion volume with Laser Capture Microdissection System PixCell II (Arcturus, ThermoFisher Scientific). We defined a perivascular area (named vascular niche, VN) as an area around blood vessel with the diagonal length two times greater than the diagonal of a particular blood vessel.

Based on above, each predefined area around blood vessels as well as the remaining lesion area (named non vascular area, non-VN) were cut out using UV laser and capture with IR laser as quickly as less than 2 hr for one slide. Dissected fragments obtained from both CCP structures of each animal were pulled as VN and non-VN, separately and RNA was extracted using PicoPure RNA Isolation Kit (ThermoFisher Scientific). The quality and integrity of the RNA was determined using the Agilent 6000 RNA Nano kit and an Agilent Bioanalyzer 2100 (Agilent). RNA samples (100 ng) representing VN and non-VN from each animal was separately amplified and biotin-labelled employing the 3'IVT express kit (Illumina), and hybridized to RatRef-12 microarrays (Illumina) – a total of 20 hybridizations, with five biological replicates per condition. The microarray experiments were conducted at the microarray facility of the Cambridge Genomic Service, Cambridge, UK according to standard

protocols. The microarray dataset has been deposited in the Gene Expression Omnibus repository, and is available with the accession number GSE93645.

## Quantitative real-time PCR

For verification of microarrays data, we used independent samples obtained using the same technique as described above. We enriched experimental conditions with the samples from intact controls (not injured, sham-operated, n = 3) and animals 14 days after toxin injection (n = 3). Total RNA from sciatic nerve and cultured cells was extracted using Tri Reagent (Sigma) combined with column isolation RNeasy Mini Kit (Qiagen). mRNA was reverse transcribed using high capacity cDNA Reverse Transcription Kit (ThermoFisher Scientific) according to the manufacturer's protocol. Quantitative real-time PCR (qPCR) reactions were performed using TaqMan Gene Expression Assays (Thermo-Fisher Scientific) with Applied Biosystems 7900HT Fast or QuantStudio 12K Flex Real-Time PCR System. We used the following primer/probe sets: Actb Rn00667869_m1; BMP4 Rn00432087_m1; CTGF Rn01537279_g1; Rn18s Rn03928990_g1; Sostdc1 Rn00596672_m1; Wnt2Rn01500736_m1; Wnt2b Rn00627297_m1.

For comparative quantification, the expression level of respective genes was normalized and expressed as a relative fold change. Delta CT values from qPCR analysis were calculated by subtracting the average value from housekeeping gene (18s RNA or Actb). The relative gene expression levels between the sample and control were determined using the formula $2^{-(S\Delta Ct - C\Delta Ct)}$ or as ratio of experimental sample/condition to intact tissue or control condition. For comparison of gene expression level in different cell types data were presented as ΔCT. Triplicate measurements were made on at least three biological replicates.

## In situ hybridisation *with cRNA probes*

Riboprobes were generated based on rat genomic sequence from NCBI database. Specific sequences of rat mRNAs were chosen and later amplified by PCR reaction using Platinum Taq DNA Polymerase High Fidelity (ThermoFisher Scientific) to avoid false incorporation of nucleotides and cloned using TopoTA Cloning Kit (ThermoFisher Scientific). The rat *BMP4* and *Sostdc1* cDNA was amplified using total RNA isolated from injured rat spinal cord (0.1% EB injection, 10 dpl). The fragment of cDNA was subcloned into plasmid pSP73 (Promega) with flanking promoters for SP6 and T7 RNA polymerase. The sequences (5'→3') of the PCR primers were: *BMP4* forward CCTGGTAACCGAA TGCTGAT, reverse CGATCGGCTAATCCTGACAT; *Sostdc1* forward GAAAGAATTCACCGCCTCAG, reverse CCTGGCTTTCACTGCAAACT. Plasmids were sequenced and transcribed in vitro to obtain antisense and sense (control) probe used for hybridization. Briefly, to label cRNA probes, following linearization of plasmids with appropriate restriction enzyme DIG labelled antisense probes were synthesized using the DIG RNA labelling kit (Roche) with suitable RNA polymerases. The target mRNA-expressing cells were visualized as a dark blue-purple deposition with NBT/BCIP–alkaline phosphatase. Double labeling with two color in situ hybridization has been described previously (*Zhao et al., 2015*). Briefly, a mixture of two different target mRNA probes labeled with DIG and FITC, respectively, was used for hybridization with the same subsequent procedure of single color in situ hybridization. After visualization of first target mRNA using NBT/BCIP, the alkaline phosphatase was inactivated by incubation of the slides at 65°C then 0.1 M glycine, pH 2.2, for 30 min, respectively. Sections were then incubated with alkaline phosphatase-conjugated antibody specific to the label of the second probe. The second target mRNA was visualized by incubating with INT/BCIP, which formed a magenta/brown deposition. Lesions were identified on digital images of sections stained with solochrome cyanine for myelin identification and the lesion area was determined using ImageJ 1.43b (http://rsb.info.nih.gov/ij/).

## Immunofluorescence

Tissue was processed as described above. Sections or cells were incubated in sodium citrate (10 mM, microwave preheated) for antigen retrieval, blocked with PBS containing 0.3% (v/v) Triton X-100 and 10% (v/v) normal donkey serum (1 hr, RT), and incubated with primary antibody overnight, 4°C.

Primary antibody used: Caspase3 R and D Systems AF835-SP; CTGF Abcam ab6992; GFAP Dako-Cytomation Z0334; GFAP BD Biosciences 556330; GFP Abcam ab6673; MPZ Abcam ab39375;

Neurofilament NF DakoCytomation M0762; Olig2 Abcam ab33427; Olig2 Millipore MABN50; Periaxin Gift from Prof. Peter Brophy, University of Edinburgh, UK; SMI71 Calbiochem NE1026; Sostdc1 Abcam 99340; Sox10 Gift from Dr. Michael Wegner, Universität Erlangen-Nürnberg, Erlangen, Germany; Sox2 Santa Cruz Biotechnology sc-17320; vWF Abcam ab6994; Iba1 Wako 019–19741, MBP R@D Systems MAB42282, tRFP Evrogen AB233.

Then, sections were incubated with appropriate AF488, AF555, or AT647- conjugated secondary antibodies (1:500, ThermoFisher Scientific). Nuclei were visualized with 4', 6'-diamidino-2-phenylindole (DAPI; 0.1 mg/ml; Sigma). Images were acquired with fluorescence microscope (Leica DM4000B) and confocal (Zeiss LSM700) microscopes.

## Statistical analysis of microarray data

Microarray data were exported from Illumina BeadStudio software, processed and normalized using the R/Bioconductor beadarray, lumi and limma packages. Before normalization, probes that were not detected (detection p value>0.01) were removed from further analysis. Remaining probes were mapped to gene identifiers from Ensembl database (gene_stable_id). For each gene, we computed a single average intensity profile from the profiles of all the probes mapped to it. The resulting average profile was then log2-transformed and used in statistical analysis and visualization. Two microarrays (derived from one animal), which did not accomplish our highest standards of analysis have been excluded from comparison. Differential expression analyses were conducted using the limma Bioconductor package. False Discovery Rate (FDR) was used to adjust for multiple hypothesis testing (*Benjamini and Hochberg, 1995*). Genes with FDR < 0.05 and with at least 1.5-fold change in expression levels were found as differentially expressed. Signalling pathways described in the Kyoto Encyclopaedia of Genes and Genomes (KEGG, 2000) were tested for overrepresentation in the list of differentially expressed genes using Fisher's exact. Differentially expressed genes between the niches were detected and volcano plot was created.

## Quantification and statistical analysis

Data analysis was performed using Prism 6 (GraphPad) software except microarray analysis. Data are represented as mean ± sd. To detect differences between experimental conditions Student's t-test or Mann-Whithney U-test were performed. p<0.05 was considered to be statistically significant. Multiple group comparisons were conducted using one-way ANOVA followed by Newman-Keuls post hoc test. In vitro assays represent three independent experiments from individual culture preparations, unless otherwise indicated.

## Acknowledgements

This work was funded by the National Science Centre grants, NN 401 584238 (MZ) and 2012/07/N/NZ3/01943 (JU-P), a programme grant from the UK Multiple Sclerosis Society (RJMF, CZ) and a core support grant from the Wellcome Trust and MRC to the Wellcome Trust – Medical Research Council Cambridge Stem Cell Institute (RJMF).

## Additional information

### Funding

| Funder | Grant reference number | Author |
| --- | --- | --- |
| National Science Centre | 2012/07/N/NZ3/01943 | Justyna Ulanska-Poutanen |
| National Multiple Sclerosis Society | Programme grant | Robin JM Franklin |
| Wellcome | Core support grant | Robin JM Franklin |
| Medical Research Council | Core support grant | Robin JM Franklin |
| National Science Centre | NN 401 584238 | Malgorzata Zawadzka |

The funders had no role in study design, data collection and interpretation, or the decision to submit the work for publication.

## Author contributions

Justyna Ulanska-Poutanen, Data curation, Formal analysis, Funding acquisition, Investigation, Visualization; Jakub Mieczkowski, Formal analysis, Investigation, Methodology; Chao Zhao, Data curation, Investigation; Katarzyna Konarzewska, Data curation, Formal analysis, Investigation, Visualization; Beata Kaza, Investigation, Methodology; Hartmut BF Pohl, Investigation, Data curation; Lukasz Bugajski, Data curation; Bozena Kaminska, Writing—original draft; Robin JM Franklin, Conceptualization, Funding acquisition, Investigation, Writing—original draft; Malgorzata Zawadzka, Conceptualization, Data curation, Formal analysis, Supervision, Funding acquisition, Investigation, Methodology, Writing—original draft. Project administration

## Author ORCIDs

Chao Zhao (iD) https://orcid.org/0000-0003-1144-1621
Hartmut BF Pohl (iD) https://orcid.org/0000-0001-6827-1665
Malgorzata Zawadzka (iD) https://orcid.org/0000-0002-3637-2709

## Ethics

Animal experimentation: All research and animal care procedures were approved by the First Warsaw Local Ethics Committee for Animal Experimentation (protocols 1020/2009, 1046/2009, 556/2014 and 431/2017) and performed according to international guidelines on the use of laboratory animals.

## Decision letter and Author response

Decision letter https://doi.org/10.7554/eLife.30325.024
Author response https://doi.org/10.7554/eLife.30325.025

# Additional files

## Supplementary files

• Transparent reporting form
DOI: https://doi.org/10.7554/eLife.30325.019

## Data availability

The microarray dataset has been deposited in the Gene Expression Omnibus repository, and is available with the accession number GSE93645.

The following dataset was generated:

| Author(s) | Year | Dataset title | Dataset URL | Database, license, and accessibility information |
|---|---|---|---|---|
| Ulanska J, Mieczkowski J, Zawadzka M | 2018 | Comparative analysis of gene expression profile of pre-defined niches within demyelinated white matter in rats | http://www.ncbi.nlm.nih.gov/geo/query/acc.cgi?acc=GSE93645 | Publicly available at the NCBI Gene Expression Omnibus (accession no: GSE93645) |

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
