## [Decision Letter]

Thank you for submitting your article "Injury-induced perivascular niche supports alternative differentiation of adult CNS progenitor cells" for consideration by *eLife*. Your article has been reviewed by three peer reviewers, and the evaluation has been overseen by a Reviewing Editor and Marianne Bronner as the Senior Editor. The reviewers have opted to remain anonymous.

The reviewers have discussed the reviews with one another and the Reviewing Editor has drafted this letter along with the Editor-in-Chief to help you consider what may be involved in satisfying the serious concerns raised in the reviews and the reviewer consultation session.

Summary:

In this manuscript, Justyna Ulanska-Poutanen and colleagues try to identify the molecular cues that cause adult OPCs to differentiate into Schwann cells or oligodendrocytes during CNS remyelination following ethidium bromide-induced demyelination in the caudal cerebellar peduncle. They argue that differentiation to Schwann cells (or better Schwann-like cells) is confined to injury-induced perivascular niches (VN) and induced by combined BMP/Wnt signals. These signals are derived from adult OPCs and endothelial cells and become dominant because of the low number of astrocytes and the low level of the astrocyte-derived dual BMP/Wnt inhibitor Sostdc1 in these areas.

The topic of the study is very interesting and has the potential to shed light on remyelination processes. It is mostly written in an intelligible way. However, there are also several major drawbacks.

Essential revisions:

The manuscript falls short of providing a molecular mechanism of how combined BMP/Wnt signals cause the fate switch in OPCs.

1) I am missing clear-cut evidence that the proposed signals really cause the fate switch in situ in the CNS. The presented data in favor of the model are first of all correlative CNS expression data. Additional support or stems from DRG co-culture experiments in which OPCs are treated with BMP4 and Wnt2b, and from experiments in which OPCs are grafted into the crushed sciatic nerve in the presence of signaling inhibitors. Neither of these experimental setups faithfully recreates CNS conditions. This is for instance evident from the fact that only Oct6 is induced in co-culture, not however any of the classical peripheral myelin genes.

2) The argument for selective BMP inhibition in the non-vascular area appears soft. As shown in Figure 3C, there appears to be, at best, a delay in expression of Sostdc1 mRNA in the vascular niche, raising the question as to what happens to the phenotype of the Schwann-like cells or ongoing myelination within that zone subsequent to the 14 day time-point. In addition, is Sostdc1 protein diffusible within the microenvironment? The effects of DORSO and USAG1 upon the numbers of myelinated segments produced by the transplanted cells in the sciatic nerve should also have been quantified, rather than just by single photomicrographs.

3) In addition, do the total numbers of Sox10 positive cells per unit area in the vascular niche at D10 and D14 post-lesion match (as superficially appear to be the case) supporting a transformation argument. Otherwise, are there other potential explanations for the reduction in Olig2 positive cells such as migration into non-vascular areas or along newly generated blood vessels?

4) It would be useful to have complementary myelin protein staining to MPZ in Figure 1F. Presumably MBP would be more widely expressed. What about PLP (an oligo specific protein); is this excluded from the perivascular niches?-if not it argues against an all or none phenomenon and/or suggests that the cells in this region are hybrids and as inconsistently conceded in the manuscript only 'Schwann cell like'.

5) Whereas the expression profile of BMP4 is increased that of BMP2 is trending down. Is further, more complete interrogation of BMP profiling/signalling required before definitive comments about its relevance can be made?

6) There will of course be some astrocytes in the perivascular area with end-feet contributing to the re-establishment of the blood-brain barrier. Is there suppression of their innate phenotype in this region or is it that these cells are fundamentally different to the reactive astrocytes in the non-vascular area?

---

## [Author Response]

Essential revisions:The manuscript falls short of providing a molecular mechanism of how combined BMP/Wnt signals cause the fate switch in OPCs.1) I am missing clear-cut evidence that the proposed signals really cause the fate switch in situ in the CNS. The presented data in favor of the model are first of all correlative CNS expression data. Additional support or stems from DRG co-culture experiments in which OPCs are treated with BMP4 and Wnt2b, and from experiments in which OPCs are grafted into the crushed sciatic nerve in the presence of signaling inhibitors. Neither of these experimental setups faithfully recreates CNS conditions. This is for instance evident from the fact that only Oct6 is induced in co-culture, not however any of the classical peripheral myelin genes.

We acknowledge that our experimental design did not focus on re-creating CNS conditions. Our specific aim was, however, to identify the causative cues that govern the alternate, PNS fate of OPC differentiation. To this end, we have primarily studied a CNS lesion model in which different OPCs differentiation fates occur in defined regions of the lesion. This led us to show that the alternate PNS fate was controlled by a perivascular microenvironment temporarily depleted of astrocyte-derived factors. Thus, a CNS environment containing reactive astrocyte supports canonical oligodendrocyte differentiation. Our logic in the current study was therefore to provide a PNS rather than a CNS environment to examine the ability of the candidate factors to induce the OPC alternative, PNS fate.

Several lines of evidence have already suggested that BMPs are key players in lineage diversification during differentiation of neural stem cells and promoting a neural crest phenotype in CNS progenitors isolated from embryonic and adult spinal cord. Our microarray data proved the importance of BMP4 and shed light on Wnt signalling as a candidate for cooperative action. Therefore, our working hypothesis was that those two pathways, normally inhibited by astrocyte-derived factor/s in the extra vascular areas of demyelinated CNS white matter are responsible for PNS fate induction.

We entirely agree that any of the classical peripheral myelin proteins were identified in our in vitro experiments (former Figure 5). We observed an occurrence of myelin segments mainly in control conditions, which suggests that the other treatments caused OPC differentiation delayed.

Therefore, in order to avoid any further confusion we propose to focus purely on in vivo studies in our revised manuscript. We now provide novel data to strengthen our conclusions and to confirm the causative role of Wnt signalling and BMP/Wnt cooperation in cell fate decision in vivo.

Namely, we have inhibited Wnt signalling in transplanted OPCs with lentiviral shRNA against Lrp6 (a Wnt co-receptor shown to be necessary for the signaling pathway induction) mimicking Sostdc1 action with regard specifically to Wnt. It has been shown that BMP4 does not interfere with Sostdc1-LRP6 binding, and functional assays showed that the ability of Sostdc1 to block Wnt activity through LRP6 is not impeded by BMP4 (Lintern et al., 2009). Now, we are able to show that inhibition of Lrp6-dependent signaling prevents OPC-to-SC differentiation even in the high BMP4 background of injured sciatic nerve. Specifically, we have infected primary OPC with lentiviral shRNA against Lrp6 and transplanted them into nerves at 1 day after crush. Defective phenotype of cells transduced with LRP6 shRNA indicates that activation of WNT signalling is required and activation of both signalling is indispensable for initiation of PNS fate differentiation which has not been proposed yet. These new data are included in a new Figure 6.

We do accept that the detailed mechanism of molecular control of the BMP/Wnt pathways cooperation in OPC alternative fate induction we describe remains unresolved. We recognize that solving this question will require for example, conditional genes tracing and OPC–specific overexpression of BMP4 and Wnt exclusively outside the vascular niche. Moreover, since we proposed Sostdc1 to be a central player to inhibit both signalling pathways the other experimental approach would be using inducible Sostdc1 knockout mice, which however, to our best knowledge, does not exist. On the other hand, rapid depletion of astrocyte, the main source of Sostdc1 protein, by which we could have reduced Sostdc1 production has been shown to cause central axonal damage.

We therefore contend that since an important milestone has been now achieved, a detailed description of the molecular control of the signalling crosstalk and their downstream elements is beyond the scope of the current study. We very much hope that the reviewer will agree with us.

2) The argument for selective BMP inhibition in the non-vascular area appears soft. As shown in Figure 3C, there appears to be, at best, a delay in expression of Sostdc1 mRNA in the vascular niche, raising the question as to what happens to the phenotype of the Schwann-like cells or ongoing myelination within that zone subsequent to the 14 day time-point. In addition, is Sostdc1 protein diffusible within the microenvironment? The effects of DORSO and USAG1 upon the numbers of myelinated segments produced by the transplanted cells in the sciatic nerve should also have been quantified, rather than just by single photomicrographs.

We apologise for not having addressed these very valid points more clearly in our original submission.

Our contention is that expression profiles of BMP4, Wnt2b and Sostdc1 are non-overlapping within defined lesion areas, and this disequilibrium, rather than BMP inhibition per se, is a key factor in determining OPC fate decision. In support of this, we have found that OPC-derived BMP4 is significantly increased in the vascular niche at 10 dpl, while expression of Sostdc1 is low at 10 dpl and limited to non-vascular niche areas. In other words, the delay of Sostdc1 expression in the vascular area allows the both signalling to be activated by their ligands present in the niche within the most critical time frame of OPC differentiation. Thus, the balance between activating and inhibitory signals and their fine-tuning affects the cell fate between 6 and 10 dpl (mostly around 10 dpl).

Sostdc1 is definitely not restricted to the cell in which it is produced, since it is secreted molecule. However, at 14 dpl, when Sostdc1 is highly expressed, equally in both compartments of the lesion, most of OPCs are already specified to either an oligodendrocyte or Schwann cell fate (according to published data and our previous results on OPC differentiation kinetics, see: Woodruff and Franklin, 1999; Zawadzka et al., 2010, see also depicted in Figure 1).

In our revised manuscript we have provided quantification of the effects of DORSO and USAG1 upon the numbers of myelinated segments produced by the transplanted cells in the host sciatic nerve, as the reviewer has requested. Moreover, we have performed experiments aimed to dissect BMP and Wnt impact on OPCs fate decision. Because Sostdc1 (or USAG1) serves as an inhibitory protein for both BMP and Wnt, our previous results might have not been fully conclusive in regard to Wnt involvement. Now we show clear evidence for Wnt involvement in induction of Schwann cell differentiation by genetic interference with Wnt signalling in differentiating OPCs (Figure 6).

3) In addition, do the total numbers of Sox10 positive cells per unit area in the vascular niche at D10 and D14 post-lesion match (as superficially appear to be the case) supporting a transformation argument. Otherwise, are there other potential explanations for the reduction in Olig2 positive cells such as migration into non-vascular areas or along newly generated blood vessels?

The total number of Sox10 positive cells per area in the vascular niche does not differ significantly between 10 and 14 dpl: detailed quantification has now been provided in the Figure 1C.

4) It would be useful to have complementary myelin protein staining to MPZ in Figure 1F. Presumably MBP would be more widely expressed. What about PLP (an oligo specific protein); is this excluded from the perivascular niches?-if not it argues against an all or none phenomenon and/or suggests that the cells in this region are hybrids and as inconsistently conceded in the manuscript only 'Schwann cell like'.

We and others already found MBP protein to be equally expressed in both CNS and PNS myelinating cells: however, we observed that MPZ and PLP are mutually exclusive within remyelinated CNS lesions. Appropriate images for *PLP* and *MPZ* mRNA identification have been now presented in Figure 1E.

Moreover, in the light of our results and those of others (Assinck at al., 2017) we use “Schwann cells” instead of “Schwann-like cells” in our manuscript.

5) Whereas the expression profile of BMP4 is increased that of BMP2 is trending down. Is further, more complete interrogation of BMP profiling/signalling required before definitive comments about its relevance can be made?

We analysed the level of expression of different members of BMP family such as BMP4, BMP2, their receptors and antagonists (noggin, gremlin, and chordin etc.) in local peri-vascular niche compared to non-vascular one. Microarray data have shown that only BMP4 was upregulated within the demyelinated lesion, regardless of the specific area examined. However, as we shown in Figure 3A, there was a significantly higher level of BMP4 in vascular areas versus non-vascular ones at 10 dpl (which we confirmed by qPCR).

The data presented in Figure 3A show the ratio of expression between vascular niche and non-VN vascular niche, and suggest low expression of BMP2 in the vascular niche. We draw the reviewers’ attention to the fact that BMP2 transcripts are detected at a very low level and the difference in expression was not significant for either comparison (i.e.: vascular niche vs. non-vascular niche and at 6 dpl vs. 10 dpl). Figure 3 has now been supported by the data on BMP2 expression added as Figure 3—figure supplement 1. The raw data for BMPs expression are presented in Author response image 1.

**Author response image 1. respfig1:** Graphs show absolute expression of *BMP4 and BMP2* in VN (green) and non-VN (red) at 6 and 10 dpl, dots represent expression level for each sample. The Y axis represents gene expression, log2; X axis – dpl.

We believe that any ambiguity can be removed by a more comprehensive description of the results and interpretation of our high-throughput data.

The logic of the microarray was to give some initial insight into how the perivascular niche and how it differs from elsewhere in the lesion. In this regard, it proved very useful and helped provide the basis for subsequent hypothesis-led experiments about some (but not all) of the potentially relevant signalling pathways that emerged. We do agree that differential expression pattern between regions primarily reflects the difference in cellular composition. Indeed, we provided in vivo and in vitro evidences that the main sources of revealed differences in gene expression are OPCs in the VN and astrocytes in the non-VN. We acknowledge that we could do more to explain why some were chosen and others (Hedgehog for example) were not and have made the necessary amendments to our revised manuscript.

6) There will of course be some astrocytes in the perivascular area with end-feet contributing to the re-establishment of the blood-brain barrier. Is there suppression of their innate phenotype in this region or is it that these cells are fundamentally different to the reactive astrocytes in the non-vascular area?

As we showed in the Figure 1—figure supplement 1, astrocytes gradually repopulate the lesion areas (confirming previously published data – e.g. Woodruff and Franklin, Glia, 1999; Gautier et al., Nature Comms, 2015). We found that at 6 dpl astrocytes were not present in the core of lesion, since they were likely killed by ethidium bromide. However, at 10 dpl some astrocytes, with processes juxtaposed to blood vessels and presumably contributing to the re-establishment of the blood-brain barrier, were occasionally observed. At 14 dpl, astrocytes are present in the vicinity of large blood vessels and by 28 dpl present throughout the entire remyelinated area (with the exception of the Schwann cell remyelinated areas) as we have now showed in the Figure 1—figure supplement 2. In such cases periaxin and GFAP were mutually exclusive, supporting our conclusion that astrocyte-derived factors inhibit the induction of OPC-to-Schwann cell differentiation within areas of the lesion in which astrocytes are present.